# A control volume finite element model for predicting the morphology of cohesive-frictional debris flow deposits

Tzu-Yin Kasha Chen[1], Ying-Chen Wu[1], Chi-Yao Hung[2], Hervé Capart[1], and Vaughan R. Voller[3]

[1]Dept of Civil Engineering and Hydrotech Research Institute, National Taiwan University, Taiwan
[2]Dept of Soil and Water Conservation, National Chung-Hsing University, Taiwan
[3]Department of Civil, Environmental, and Geo- Engineering, University of Minnesota, USA

**Correspondence:** Tzu-Yin Kasha Chen (d06521004@ntu.edu.tw)

**Abstract.** To predict the morphology of debris flow deposits, a control volume finite element model (CVFEM) is proposed, balancing material fluxes over irregular control volumes. Locally, the magnitude of these fluxes is taken proportional to the difference between the surface slope and a critical slope, dependent on the thickness of the flow layer. For the critical slope, a Mohr–Coulomb (cohesive-frictional) constitutive relation is assumed, combining a yield stress with a friction angle. To verify the proposed framework, the CVFEM numerical algorithm is first applied to idealized geometries, for which analytical solutions are available. The Mohr–Coulomb constitutive relation is then checked against debris flow deposit profiles measured in the field. Finally, CVFEM simulations are compared with laboratory experiments for various complex geometries, including canyon-plain and canyon-valley transitions. The results demonstrate the capability of the proposed model and clarify the influence of friction angle and yield stress on deposit morphology. Features shared by the field, laboratory, and simulation results include the formation of steep snouts along lobe margins.

## 1 Introduction

When they transition from steep gullies to milder topography, debris flows typically spread out and slow down to form fresh deposits. By burying houses, bridges, or other assets, these may cause considerable damage to communities and infrastructure (Liu and Huang, 2006; Scheidl et al., 2008; Tai et al., 2019). This is illustrated in Fig. 1 for a case in Taiwan (courtesy of the Chi Po-lin Foundation, 2009), where debris flow deposition near a gully mouth buried the lower stories of multiple buildings. To mitigate debris flow hazards, it is therefore important to anticipate the possible extent and thickness of their deposits.

To simulate the flow and deposition of debris flows, many highly resolved models have been proposed. These typically apply mass and momentum balance equations to flows over non-erodible (O'Brien et al., 1993; O'Brien, 2006; Liu and Huang, 2006; Murillo and García-Navarro, 2012; Pudasaini, 2012; Kowalski and McElwaine, 2013; Gregoretti et al., 2016; Meng and Wang, 2016; Tai et al., 2019; Pudasaini and Fischer, 2020) or erodible substrates (Armanini et al., 2009; Bartelt et al., 2017). Such simulations, however, require detailed hydrological input data and various rheological parameters which may be difficult to obtain, and may also differ dramatically from one case to another. In this context, it is worth exploring whether reduced complexity models could predict key features of debris flow deposits with less computational effort and more limited data

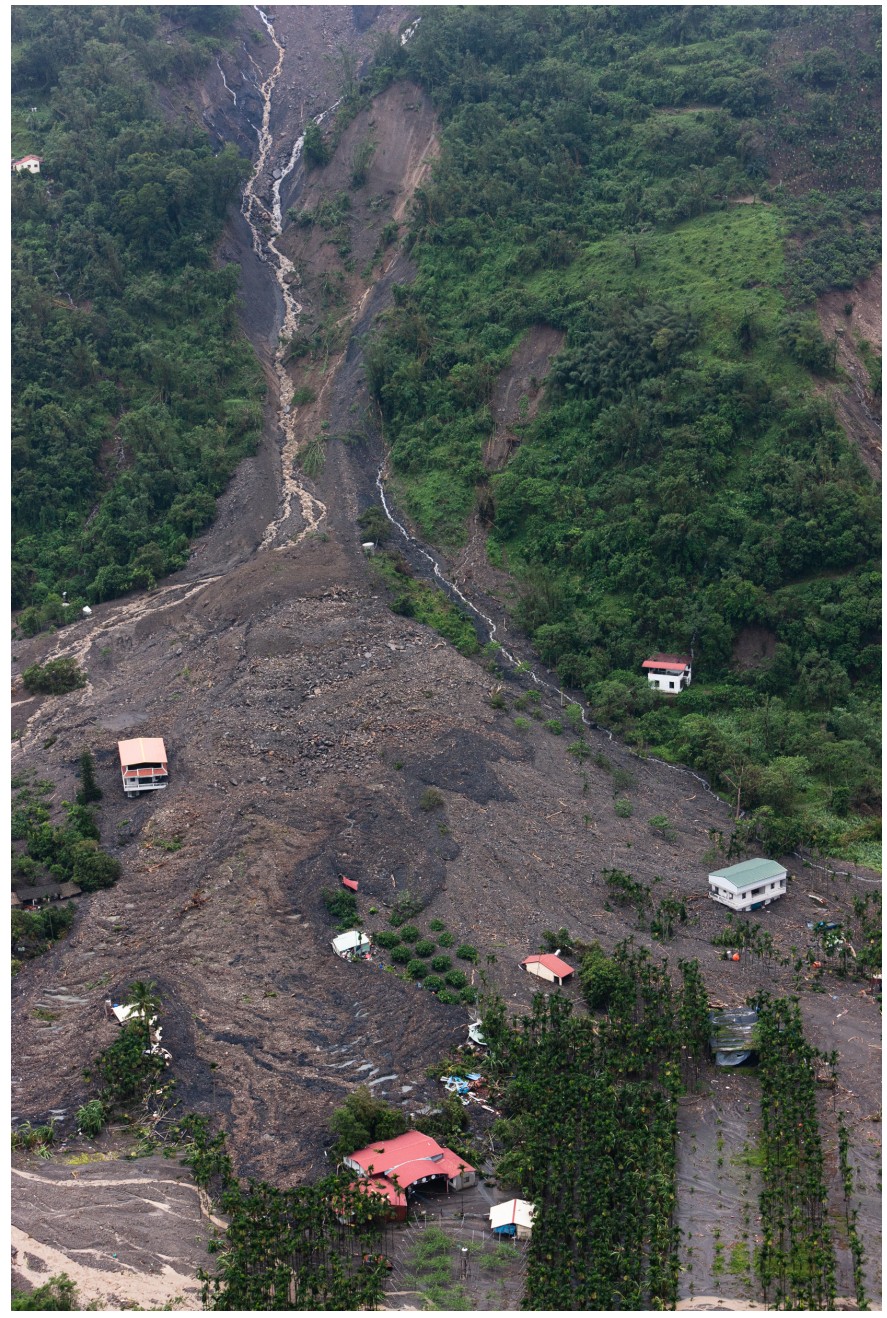

**Figure 1.** Aerial view of the debris flow deposit formed at Xinfa, Southern Taiwan, during Typhoon Morakot in August 2009 (Photograph by Chi Po-lin. Provided by Chi Po-lin Foundation © Above Taiwan Cinema, Inc. Only authorized for this article and cannot be extracted, distributed, or reproduced separately in any form. Offenders shall be prosecuted for legal responsibility by relevant laws and regulations.)

requirements. Unlike existing models that also attempt to predict the runout at high velocities, we limit our scope and focus on predicting the final deposit morphologies of debris flows, modelled as slow, quasi-static processes.

A class of reduced complexity models developed for fluvial problems rests on defining a constitutive model for the mass flux, which in turn can be used with a mass balance equation (e.g., the Exner equation) to evolve the bed surface elevation. For applications to alluvial fans and river deltas, for instance, some models have been proposed that simply set the mass flux proportional to the current slope at that point (Voller and Paola, 2010; Lorenzo-Trueba and Voller, 2010; Lorenzo-Trueba et al., 2013). More sophisticated approaches employ the device of a critical threshold (Mitchell, 2006; Lai and Capart, 2007), whereby sediment transport occurs only when the bed inclination exceeds a critical slope (Lai and Capart, 2007; Hsu and Capart, 2008; Lai and Capart, 2009). In these models, the critical slope for the fluvial sediment flux can be derived by considering the friction stress at the sediment-water interface (the Shields stress). In some sense, this idea of a critical slope is analogous to the angle of repose governing the shapes of dry sand piles (Kuster and Gremaud, 2006; Giudice et al., 2019), or the morphology of idealized deltas and fans (Ke and Capart, 2015; Zhao et al., 2019; Chen et al., 2022).

Mass flux models have also been used to model mud flows. In particular, we refer to the work of Yuhi and Mei (2004) where a flux law was obtained by combining lubrication theory with a cohesive yield stress criteria. Predictions from this model were verified by comparing with analytical solutions which constrain the slope of the deposit, in axi-symmetric domains, based on a cohesive yield stress criteria (Coussot et al., 1996; Yuhi and Mei, 2004). Unlike what might be seen in a sand pile or fluvial system close to the threshold, here the slope at a point varies with the thickness of the deposit.

Contrasting with fluvial and mud flows, for debris flows it is believed that both friction angle and yield stress can affect the morphology of deposits (O'Brien et al., 1993; Mangeney et al., 2010; Murillo and García-Navarro, 2012; Pudasaini, 2012; Gregoretti et al., 2016; Tai et al., 2019; Pudasaini and Fischer, 2020). The study of Coussot et al. (1996) emphasizes this point. Using only a yield stress criterion, these authors derived solutions for deposit profiles which they compared with surveyed debris flow transects. This model was found to work well for cohesive debris flow deposits with high clay content. For lower clay content, however, deposit inclinations are more consistent with control by the saturated angle of friction (Takahashi, 1991). For debris deposits mixing coarse and fine material, therefore, it appears necessary to consider both a yield stress and a saturated friction angle, as in the well-known Mohr–Coulomb model for cohesive-frictional materials.

The objective of the current work is 3-fold, first, we will develop a mass flux expression that considers both friction angle and yield stress in setting the critical slope under a quasi-static assumption. Secondly, we will use this mass flux in an unstructured control volume finite element method (CVFEM) solution of the Exner mass balance equation to arrive at, for a given input mass, predictions of the final deposit location and shape. Finally, we will assess the predictive performance of this model by comparing predictions with available closed-form expressions, experimental measurements, and field observations.

In line with our objectives, we note that, in general, alluvial and debris fans build up over time in more complex ways than those immediately addressed by our proposed model and experiments. For example, channel formation, migration, and avulsion are expected to significantly affect fan evolution, especially for large scale debris flow fans. For alluvial fan experiments devoted to these processes, the reader is referred to Le Hooke and Rohrer (1979), Whipple et al. (1998), Delorme et al. (2018), and Savi et al. (2020). Our focus here, however, is on the formation of fresh deposits, possibly over a pre-existing fan surface, by

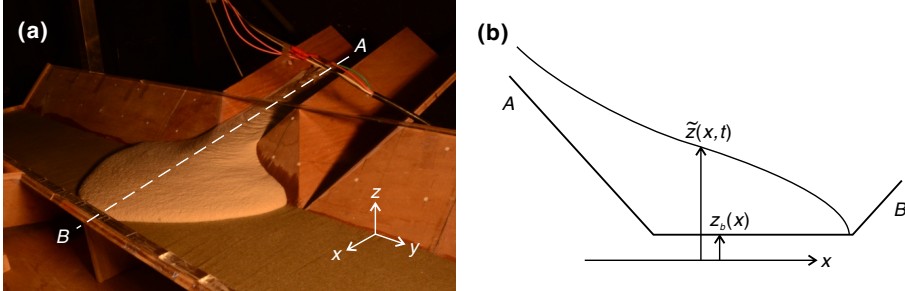

**Figure 2.** Deposition of a cohesive-frictional material over a substrate of known geometry. (a) Experimental case featuring a symmetric deposit; (b) Schematic section along the deposit centerline.

unchannelized debris flows. For such conditions, illustrated by Fig. 1, we hope to formulate and verify a simplified model that could later be extended to more general conditions.

The paper is structured as follows. Section 2 presents the governing equations that form the core of our model. The CVFEM algorithm developed to obtain numerical solutions is then described in Sect. 3. Section 4 describes how we incorporate a Mohr–Coulomb constitutive relation into this framework. In Sect. 5, we explain how to supplement our CVFEM with a flux limiter, to model flow over non-erodible surfaces. In Sect. 6, we check simulations against available analytical solutions. In Sect. 7, we verify our model by comparing results with field data and laboratory experiments. Finally, in Sect. 8, we discuss the contribution and limitations of our work, emphasizing how our model can help understand the influence of material properties on the morphology of debris flow deposits.

## 2   Governing equations

To write governing equations, we consider a debris mixture depositing over a fixed substrate of arbitrary topography. An example is shown in Fig. 2a: supplied upstream of a steep triangular channel, the mixture flows into a trapezoidal channel of mild inclination, where it spreads out and slows to a complete stop. We denote by $\tilde{z}(x,y,t)$ the time-varying surface elevation during flow, and by $z_b(x,y)$ the underlying bed topography. The corresponding profiles are shown on Fig. 2b on a schematic section.

To capture the deposition process and predict the final deposit morphology, we express mass conservation by the Exner equation (Exner, 1920, 1925)

$$\frac{\partial \tilde{z}}{\partial t} = -\nabla \cdot \mathbf{q} + Q_{in}\delta(\mathbf{x_s}), \tag{1}$$

where $\mathbf{q} = (q_x, q_y)$ is the volumetric flux (volume transferred per unit width and time), $\nabla \cdot \mathbf{q}$ with $\nabla = (\partial/\partial x, \partial/\partial y)$ is the divergence of this flux, $\delta$ is the Dirac delta function, $\mathbf{x_s} = (x_s, y_s)$ is the location of the source, and $Q_{in}$ is the inflow source volumetric flux. For simplicity, we assume that the flow is sufficiently slow to be regarded as quasi-static, allowing inertia effects to be neglected. At each location $(x, y)$, the flux $\mathbf{q}$ is assumed aligned with the direction of steepest descent according

to

$$\mathbf{q} = -\nu \nabla \tilde{z}. \tag{2}$$

The diffusivity $\nu$, however, is not assumed constant but instead depends on the local surface slope $||\nabla \tilde{z}||$ according to the formula

$$\nu = \nu^* \max\left( \frac{||\nabla \tilde{z}|| - S_c}{||\nabla \tilde{z}||}, 0 \right). \tag{3}$$

where $\nu^*$ is a real and positive constant, and $S_c(x,y)$ is a critical slope dependent on material properties and on the local instantaneous thickness of the flow layer. This dependence of $S_c$ on the flow layer thickness is derived in Sect. 4. Combining Eqs. (1), (2) and (3), we see that we obtain a non-linear diffusion process with a diffusivity $\nu$ that depends on the difference between the magnitude of the local gradient and the critical slope $S_c$. With this model, the flux is only non-zero when the local slope $||\nabla \tilde{z}||$ exceeds the critical slope $S_c$. By contrast, models that consider momentum effects can produce local deposit slopes that are smaller than critical slopes (e.g. Tregaskis et al., 2022). In our model, on the deposit surface where the flow slows down to a complete stop, the flux $\mathbf{q}$ vanishes as the local slope $||\nabla \tilde{z}||$ decreases from a value that exceeds the critical slope $S_c$ to exactly the critical slope $S_c$, imposing the mathematical condition that

$$||\nabla \tilde{z}|| = S_c \tag{4}$$

everywhere on the final deposit surface. Make particular note that the critical slope developed in our model (see Sect. 4) will involve the sum of two components, a constant friction slope and a yield stress term that will be an inverse function of the deposits thickness, thus the final slope over the predicted deposited debris flow may not take a constant value.

To incorporate the above flux definition (Eqs. 2 and 3) in an Exner balance, our model includes three main components. First, we need a numerical method to solve the governing mass balance equation with the proposed flux model. Second, we need to derive an appropriate expression for the critical slope—in doing this we will consider both a friction angle and a yield stress. Third, we need to provide a limiter in our evolution algorithm to avoid fluxing out from a control volume more than the amount of material available.

## 3 Numerical method

To solve the Exner equation as formulated above, we adopt the control volume finite element method (CVFEM), a method first proposed by Winslow (1966) and later extended by Baliga and Patankar (1980, 1983), Voller (2009) and Tombarevic et al. (2013). The CVFEM is a useful tool for this application because it couples the finite element flexibility of fitting the domain geometry with the explicit mass balance of the control volume.

The application of the CVFEM to model debris flow deposits over an existing topography starts by identifying a 2-D planar problem domain $(x,y)$ and then covering this domain with a mesh of connected, non-overlapping, plane geometric elements. In our case, we use a rectangular domain and cover it with an unstructured mesh of linear triangle elements (Fig. 3a). Each triangular element is associated with three vertex node points (locally labeled $A, B$, and $C$) (Fig. 3b). This will result in

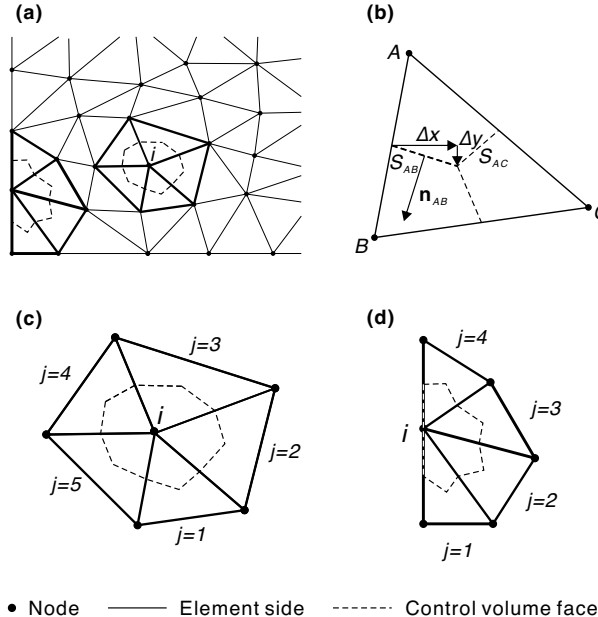

**Figure 3.** Global and local mesh geometry: (a) The discretized domain and elements; (b) a triangular element divided by the segments connecting the centroid and the midpoint of each side; (c) the control volume and the region of support of an internal node; (d) those of a node on the boundary.

$i = 1, 2, \ldots N$ node points in the domain, each storing values for the fixed bed substrate elevations $z_b(x, y)$, assumed given, and for the time-dependent flow surface elevations $\tilde{z}(x, y, t)$, to be determined. To evaluate the values of $z_b$ and $\tilde{z}$ at internal points in an element we use the classic finite-element interpolation based on linear shape functions. In this way, at a point $(x, y)$ in a
115 given element we approximate the bed substrate elevation as

$$z_b(x, y) = n_A(x, y) z_{b_A} + n_B(x, y) z_{b_B} + n_C(x, y) z_{b_C} \tag{5}$$

and the flow surface elevation as

$$\tilde{z}(x, y, t) = n_A(x, y) \tilde{z}_A(t) + n_B(x, y) \tilde{z}_B(t) + n_C(x, y) \tilde{z}_C(t), \tag{6}$$

where the shape functions, $n_A, n_B$ and $n_C$, linear functions in $x$ and $y$, take a unit value at nodes $A, B$ and $C$ respectively and
120 vanish along the element sides opposite the labeled node, i.e, sides $B - C$, $C - A$, and $A - B$ respectively. Thus, the CVFEM discretization provides piece-wise linear approximations of the bed substrate and flow surfaces. In particular, we note that in any element $j$ in our domain we can readily approximate the surface gradient by

$$\nabla \tilde{z}_j = \left( n_{A_x} \tilde{z}_A + n_{B_x} \tilde{z}_B + n_{C_x} \tilde{z}_C, n_{A_y} \tilde{z}_A + n_{B_y} \tilde{z}_B + n_{C_y} \tilde{z}_C \right), \tag{7}$$

where, $n_{A_x}, n_{A_y}$ etc are the derivatives of the shape functions. Due to the linear nature of the shape functions, we note this
approximation renders a constant value for the slope in each element.

To move on, we construct an additional geometric element on our grid of triangular elements. We join the midpoint of each element side to the centroid of each element, generating a set of connected non-overlapping control volumes around each node $i$ in the domain, see Fig. 3c,d. Thus the control volume around node $i$ has $j = 1, 2, \ldots m$ elements connected to it (the region of support), and each of these elements contains two faces of the control volume. To discretize our governing equation, Eq. (1), we integrate the equation over the control volume, use the divergence theorem, and make an explicit finite difference approximation in time to arrive at a discrete equation for the surface elevation at each node point and time step,

$$\frac{\tilde{z}_i^{\text{new}} - \tilde{z}_i}{\Delta t} = -\frac{1}{A_{CV,i}} \sum_{j=1}^{m} Q_j + \frac{Q_{in,i}}{A_{CV,i}}, \tag{8}$$

where $A_{CV,i}$ is the area of the control volume, $Q_{in,i}$ is the source flux at node $i$ and

$$Q_j = \int_{S_{AB}+S_{AC}} \mathbf{q}_j \cdot \hat{\mathbf{n}} \, ds \tag{9}$$

is the net discharge out of the control volume across the two faces in element $j$, e.g., sides $S_{AB}$ and $S_{AC}$ in Fig. 3b.

With an appropriate constitutive equation for determining the critical slope–see discussion below–we can use our approximations for the deposit slope in the element, Eq. (7) to, through Eq. (2), arrive at an approximation for the flux $\mathbf{q}_j = (q_{x_j}, q_{y_j})$ in element $j$; we should expect this value to be constant over the element. Further, if we use $\Delta x$ and $\Delta y$ to express the change in the $x$ and $y$ values along a face as we move counter-clockwise around node $i$ (see Fig. 3b), we can express the constant outward normal on a face with length $\ell$ as $\mathbf{n} = (\Delta y/\ell, -\Delta x/\ell)$. This provides us enough information to fully approximate the discharge in Eq. (9) in terms of the current nodal values of $\tilde{z}_i$ in the element (for full details refer to Voller (2009)). On making this approximation for each element in the support of node $i$ and rearranging Eq. (8) we arrive at the following update for the surface elevation:

$$\tilde{z}_i^{\text{new}} = \tilde{z}_i - \frac{\Delta t}{A_{CV,i}} \left( \sum_{j=1}^{m} Q_j - Q_{in,i} \right), \tag{10}$$

We note that when a node $i$ is on the domain boundary, see Fig. 3d, we set the discharge across the control volume faces that coincide with the boundary to zero. Hence Eq. (10) provides us with an explicit means of updating the nodal values of the surface elevation at time $t + \Delta t$ from the known values at time $t$. To speed up computations yet insure numerical stability, we use a dynamic time step $\Delta t = 0.2\Delta \ell^2 / \max(\nu_{ele})$, where $\Delta \ell$ is the average element size, and $\nu_{ele}$ is the element diffusivity given by $\nu_{ele} = \nu^*(||\nabla \tilde{z}_{ele}|| - S_{c,ele})/||\nabla \tilde{z}_{ele}||$. To let material diffuse rapidly to surrounding elements when the element slope exceeds the critical slope, we set $\nu^* = 100 \max(Q_{in})$.

## 4 Critical slope

In the previous sections, we assumed that flow occurs when the surface slope exceeds a critical slope, or, upon assuming that the direction of steepest descent coincides with the $x$-axis

$$\left| \frac{\partial \tilde{z}}{\partial x} \right| > S_c, \tag{11}$$

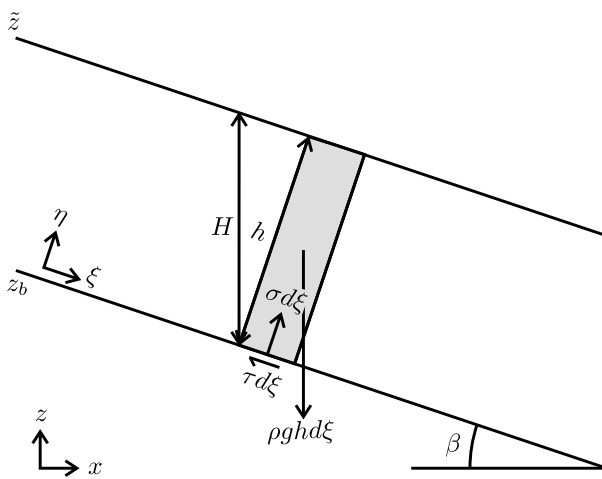

**Figure 4.** Force balance of a small piece of material on a fixed bed whose local gradient has a value equal to $\tan\beta$ and direction pointing towards $\xi$.

To set this critical slope, we adopt a Mohr–Coulomb failure criterion. For flow to occur, the shear stress $\tau$ at the base must then satisfy

$$\tau > \sigma\tan\phi + \tau_Y \,, \tag{12}$$

where $\sigma$ is the normal stress, $\phi$ is the saturated friction angle dependent on the solid fraction, the void fraction, and the fine content in the fluid (Takahashi, 1991), and $\tau_Y$ is the yield stress. When the deposit surface slope is less than or equal to the critical slope (i.e., $|\partial\tilde{z}/\partial x| \leq S_c$), the mixture remains in static equilibrium, with $\tau \leq \sigma\tan\phi + \tau_Y$. In the limiting state, we can therefore use a force balance to derive an expression for the critical slope.

In the CVFEM model, we express this force balance element by element under the following two simplifying assumptions: (i) the surface slope in an element is uniform (a direct consequence of our choice of linear elements); (ii) the flow thickness in an element is also uniform. This latter restriction is needed to keep expressions simple, but will still allow us to apply the model to flows of variable thickness. Under these assumptions, we can simply consider a 2-dimensional force balance in the $(\xi, \eta)$ coordinate system aligned with the surface inclination, as illustrated in Fig. 4. Force balance in the normal and tangential directions can then be expressed as

$$\sigma d\xi = \rho gh\cos\beta\, d\xi \,, \quad \tau d\xi = \rho gh\sin\beta\, d\xi \,, \tag{13}$$

where $\rho$ is the density of the mixture, $g$ the gravitational acceleration, $h$ the oblique layer thickness in the $\eta$ direction, and $\beta$ the bed inclination angle. To move forward, we note, by our assumptions, that

$$\frac{\partial\tilde{z}}{\partial x} = \tan\beta \,. \tag{14}$$

and that the vertical and oblique thicknesses are related by

$$H = \frac{h}{\cos\beta} \,. \tag{15}$$

Thus, on substituting Eqs. (14) and (15) into the force balance relations, Eq. (13), we obtain the following expression for the shear stress

$$\tau = \rho g h \sin\beta = -\rho g H \frac{\partial \tilde{z}}{\partial x}. \tag{16}$$

an expression that matches the derivation made by Yuhi and Mei (2004). Finally, on substituting this shear stress into the Mohr–Coulomb criterion, we arrive at a model for the critical slope

$$S_c = \left| \frac{\partial \tilde{z}}{\partial x} \right|_{\max} \approx \frac{\tau}{\rho g H} \approx \tan\phi + \frac{\tau_Y}{\rho g H}. \tag{17}$$

The critical slope in each element can therefore be determined by setting values for the saturated friction angle and yield stress, taking into account the local vertical layer depth $H = \tilde{z} - z_b$. What distinguishes our expression from previous suggestions for the critical slope Liu and Mei (1989), Coussot et al. (1996) and Yuhi and Mei (2004) is the appearance of the friction angle in addition to the yield stress. We emphasize that this combination of the friction angle and bed thickness in the definition of the critical slope is an essential ingredient in our model. This affects the slope of the final deposit as follows. Where the thickness of the deposit is large, for instance close to the source, the final slope will approach the constant value $\tan\phi$. Towards the margins, by contrast, where the deposit thickness decreases, steeper slopes and snout-like features will produced, consistent with observations.

## 5 Flux limiter

In our CVFEM model, we assume a non-eroding bed substrate. This will require the use of a "flux limiter" to ensure mass conservation in an element over each time step of the calculation. Over a time step, we cannot flux out more material than what is available at the beginning of the time step.

With reference to the selected element in Fig. 3b, we note that one-third of the element area $A_{ABC}$ contributes to the control volume around node $A$ and thus, at the start of a time step, the material available for fluxing from this sub-section of the control volume will be $\frac{1}{3}(\tilde{z}_A - z_{bA})A_{ABC}$. In this way, over a time step $\Delta t$, the maximum discharge that can be fluxed out from this section, contributing to the inflows to nodes $B$ and $C$, is given by

$$Q_{\max,A} = \frac{\tilde{z}_A - z_{bA}}{\Delta t} \frac{A_{ABC}}{3} \tag{18}$$

From this, following the time step calculation of the flux $Q_A$ across faces $S_{AB}$ and $S_{AC}$, we can provide a limiter by setting

$$Q_A = C_A Q_A \tag{19}$$

where the limiting factor $\leq 1$ is calculated as

$$C_A = \begin{cases} Q_{\max,A}/Q_A, & \text{if } Q_A > Q_{\max,A} \\ 1, & \text{otherwise.} \end{cases} \tag{20}$$

Similar limiters must likewise be applied to the outflows from nodes $B$ and $C$. In practice, to ensure that fluxes balance out, we apply a single value of the limiting factor

$$C = \min(C_A, C_B, C_C) \tag{21}$$

to each element in the solution domain.

## 6 Analytical solutions

As the flow spreads and slows, it will eventually come to a complete stop and freeze in place. At each point of the resulting deposit, the limit equilibrium condition, Eq. (4), will then be satisfied. If, say because of symmetry, the surface gradient along a certain transect is everywhere aligned with this transect, then the surface profile will satisfy the simpler equation

$$\frac{\partial \tilde{z}}{\partial x} = \pm S_c = \pm \tan\phi \pm \frac{\tau_Y}{\rho g H}, \tag{22}$$

with coordinate $x$ taken along the transect direction. In this expression, the plus operators denote downhill deposition ($\tilde{z}$ and $z_b$ decreasing in the same direction), and the minus operators denote uphill deposition ($\tilde{z}$ and $z_b$ decreasing in opposite directions). Substituting $\tilde{z} = z_b + H$, the equation becomes an ODE for the deposit thickness

$$\frac{\partial H}{\partial x} = -\frac{\partial z_b}{\partial x} + \frac{\partial \tilde{z}}{\partial x} = -\tan\beta \pm \tan\phi \pm \frac{\tau_Y}{\rho g H}. \tag{23}$$

For the special case in which the bed slope $\partial z_b/\partial x = \tan\beta$ is constant, Eq. (23) becomes a first-order autonomous ODE that can be integrated analytically. In implicit form, the resulting depth profile $H(x)$ is given by

$$x - x_0 = \begin{cases} (H(x) - H(x_0))/A & \text{if } B = 0, \\ (H(x)^2 - H(x_0)^2)/(2B) & \text{if } A = 0, \\ (AH(x) - B\ln(|AH(x) + B|))/A^2 - C & \text{otherwise,} \end{cases} \tag{24}$$

where

$$A = -\tan\beta \pm \tan\phi, \quad B = \pm\frac{\tau_Y}{\rho g}, \quad C = \frac{AH(x_0) - B\ln(|AH(x_0) + B|)}{A^2}. \tag{25}$$

In the above expressions, $H(x_0)$ is the boundary condition at $x_0$, which can be any point within the depositing region. Note that $A$ will be zero for frictionless material deposits on a horizontal plane, or frictional materials depositing downhill when the friction slope equals the bed slope, and $B$ will be zero when there is no yield stress. In what follows, these analytical solutions will be used for three purposes: clarify model properties, verify the numerical method, and calibrate material parameters when comparing model results with field and laboratory data.

As an example, analytical solutions for the centerline profiles of cohesive-frictional deposits over an inclined plane are illustrated in Fig. 5. For each case and deposit height $H$, we supply material at a single point corresponding to the apex of each deposit. To facilitate comparison, the source locations are adjusted to let the deposits have the same toe location. These locations $x_s$ are determined using the formula $x_s = (AH - B\ln(|AH + B|))/A^2 - C$, where $A = -\tan\beta + \tan\phi$, $B =$

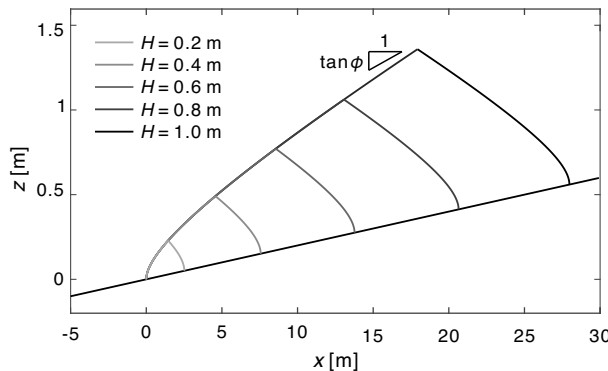

**Figure 5.** Analytical solutions for the centerline profiles of cohesive-frictional deposits on an inclined plane of slope $\tan\beta = 0.02$, for different deposit heights, assuming identical material properties $\tan\phi = 0.05$, $\tau_Y/(\rho g) = 0.01$ m.

$\tau_Y/(\rho g)$, $\quad C = (-B\ln(B))/A^2$. In all cases, the material properties are the same, and the origin is taken at the downstream end of each deposit. This representation is chosen to highlight two important features of the solutions. First, the shape of the deposit toe does not change with the size of the deposit, and depends only on the bed slope and material properties. Secondly, the different material properties affect separate features of the profiles. The yield stress $\tau_Y$ controls the scale of the steep snouts, where the deposit thickness reaches zero, whereas the friction slope $\tan\phi$ sets the deposit inclination far away from the snouts, where the deposit thickness becomes large.

It follows from these properties that a single profile of sufficient length through the toe of a deposit is sufficient to calibrate the material properties of the model. This is very useful as it greatly facilitates model application to field and experimental cases. A second implication is that, for deposits of large size compared to the scale of the snouts, deposit shapes may be well approximated by surfaces of constant slope. For the deposits of Fig. 5, setting the yield stress to zero would produce upright cones of slope $\tan\phi$ centred at the apex of each deposit. In general, however, the morphology of deposits will be affected by both the yield stress and the friction angle.

## 7 Numerical model evaluation

In this section, we evaluate the CVFEM numerical model by comparing results with analytical solutions. This provides an opportunity to show how model results depend on material parameters, for some additional simple cases. We also examine how mesh geometry and size affect the accuracy and performance of the model.

### 7.1 Comparison with analytical solutions

To verify our CVFEM algorithm we consider deposits formed by supplying material from a point source onto three idealized geometries: (i) a horizontal plane, (ii) an axisymmetric conical basin of slope $\tan\beta = 0.05$, and (iii) an inclined plane of constant slope ($\tan\beta = 0.02$). The CVFEM model for each of these cases operates in Cartesian coordinates and will produce

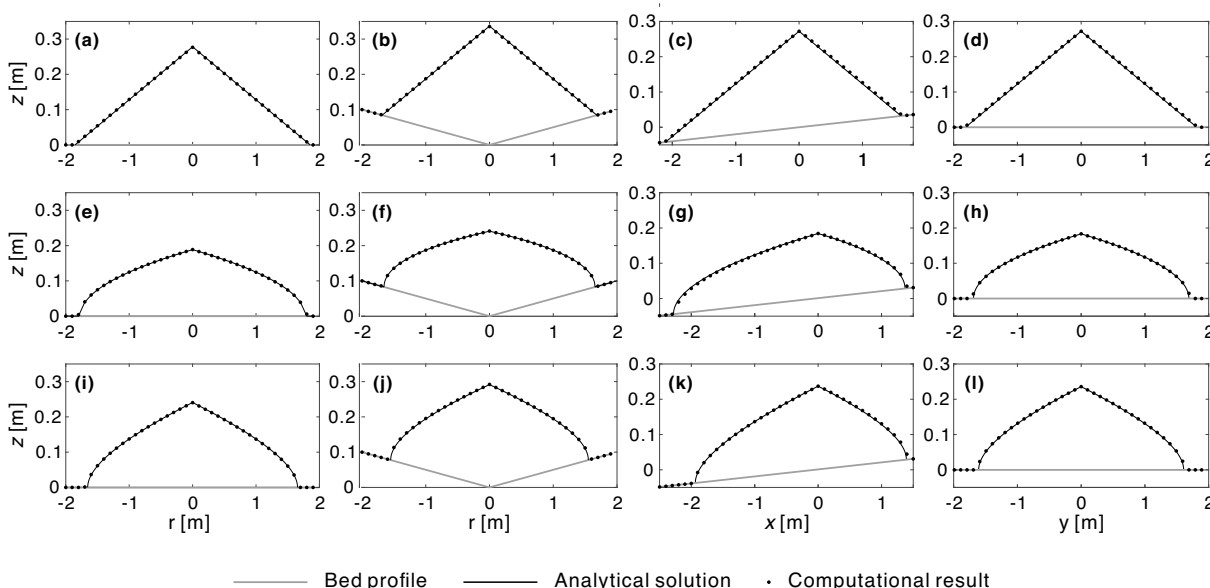

**Figure 6.** Comparison between computational and analytical solutions for different material parameters and geometries: (a,e,i) radial deposit profiles on horizontal plane; (b,f,j) radial deposit profiles on conical basin; (c,g,k) longitudinal deposit profiles on inclined plane; (d,h,l) transverse deposit profiles on inclined plane; (a-d) deposit with friction angle and no yield ($\tan\phi = 0.15$, $\tau_Y/(\rho g) = 0$ m); (e-h): deposit with yield stress and no friction angle ($\tan\phi = 0$, $\tau_Y/(\rho g) = 0.01$ m); (i-l): deposit with both friction angle and yield stress ($\tan\phi = 0.05$, $\tau_Y/(\rho g) = 0.01$ m).

3D deposit shapes. Thus, to compare with analytical solution profiles we need to select appropriate transects. For the horizontal plane and conical basin cases, we examine radial profiles (see Fig. 6a,b,e,f,i,j). For the inclined plane, we select two profiles through the source point: a longitudinal profile in the direction of the base slope, and a transverse profile orthogonal to this direction (see Fig. 6c,d,g,h,k,l). For the longitudinal profile (Fig. 6c,g,k), we can use the analytical solution in Eq. (24) as the exact solution. For the transverse profile (Fig. 6d,h,l), the transect is not a true symmetry axis. Nevertheless, the analytical solution obtained by setting $\tan\beta = 0$ can be used as an approximate solution. For each case, we impose a fixed thickness of the deposit at the origin for both analytical solutions and numerical solutions.

To show how parameters affect results and check the numerical model under different assumptions, we compare numerical and analytical solutions for three groups of material properties: (1) $\tan\phi > 0$, $\tau_Y = 0$, (2) $\tan\phi = 0$, $\tau_Y > 0$, and (3) $\tan\phi > 0$, $\tau_Y > 0$. We find excellent agreement between the computational results and the analytical solutions in each cases regardless of the choice of parameters (Fig. 6) and therefore verify the proposed CVFEM algorithm.

In the cases with constant friction stress and no yield ($\tan\phi > 0$, $\tau_Y = 0$, Fig. 6a–d), the simulated final deposits have constant surface slopes equal to the friction slope, which is consistent with physical and computational models for sand piles (Kuster and Gremaud, 2006; Giudice et al., 2019).

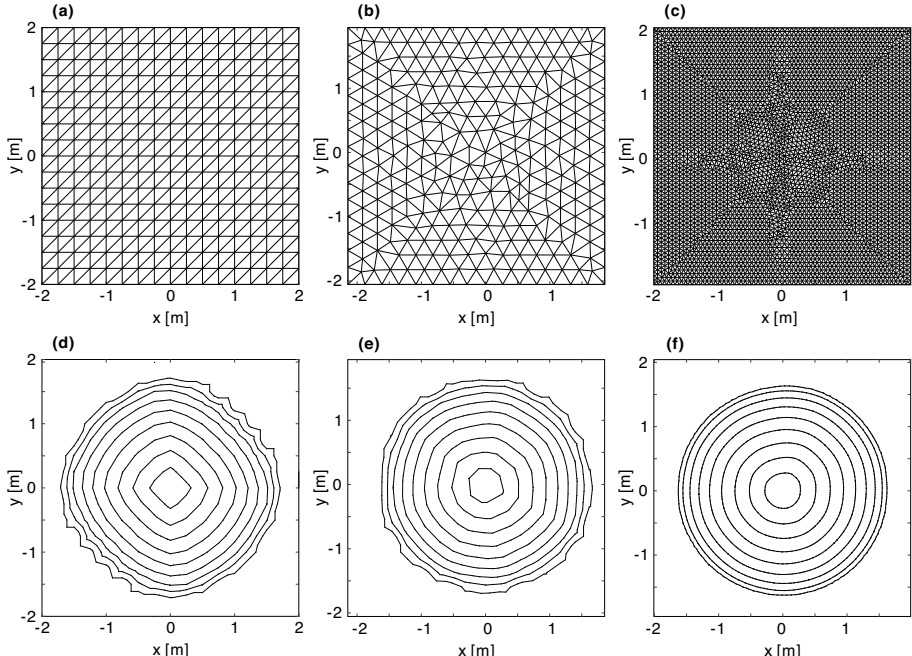

**Figure 7.** Mesh geometries (a,b,c) and calculated contours (d,e,f) for the deposition of a prescribed volume of material on a horizontal substrate: (a,d) structured mesh; (b,e) unstructured mesh; (c,f) fine unstructured mesh (8408 elements). The contours show deposit elevations $\tilde{z} = 0.1h, 0.2h, \ldots, 0.9h$.

In the cases with only yield stress ($\tan\phi = 0$, $\tau_Y > 0$, Fig. 6e–h), we obtain piles with mild slopes in the central regions and steep slopes along the margins of the deposit, resulting in toes that have a snout-like profile. This matches the analytical solutions and models proposed by Coussot et al. (1996) and Yuhi and Mei (2004) for slow mud flows (fluids with a Bingham plastic rheology). By considering the yield stress, it is therefore possible to reproduce the snout-like toes observed along the margins of many debris flow, mud flow and snow avalanche deposits (Johnson, 1970; Pudasaini and Hutter, 2007).

Finally, in the cases with both friction and yield stress ($\tan\phi > 0$, $\tau_Y > 0$, Fig. 6i–l), we note that snout-like profiles are again obtained at the toes. Away from the toes, however, the deposit slope now tends toward a finite inclination, controlled by the friction angle. Overall the results in Fig. 6 clearly demonstrate how the friction angle and yield stress affect deposit shapes.

## 7.2 Influence of mesh geometry and size

By using triangular elements as building blocks, the CVFEM model can be applied to either structured or unstructured meshes. In Fig. 7, we show how model results are affected by mesh geometry and size. For these calculations, we again consider a simple test case in which material supplied at the origin deposits over a horizontal substrate, under the combined influence of friction angle and yield stress ($\tan\phi > 0$, $\tau_Y > 0$). For these tests a prescribed volume of material is supplied, by controlling the accumulated discharge supplied at the source.

**Table 1.** Influence of mesh size on model accuracy and computational time

| Avg. element size [m] | # of elements | $(h-H)/h$ | $(R_{10\,\mathrm{max}} - R_{10\,\mathrm{min}})/r_{10}$ | Computational time [s] |
|---|---|---|---|---|
| 0.265 | 526 | 0.063 | 0.092 | 0.092 |
| 0.132 | 2116 | 0.032 | 0.037 | 2.46 |
| 0.066 | 8612 | 0.020 | 0.012 | 46.5 |
| 0.033 | 33986 | 0.011 | 0.007 | 1117.4 |

Three different meshes are considered: a structured mesh, built from triangular elements laid out in a row-column pattern (Fig. 7a); an unstructured mesh, constructed by the mesh generation algorithm of Engwirda (2014) (Fig. 7b); a fine unstructured mesh, constructed by the same algorithm (Fig. 7c). The corresponding model results are shown in Fig. 7d–f, representing the calculated topography by elevation contours.

In Fig. 7d, clear directional errors can be seen when results are computed on the structured mesh. In this case, the deposits contours visibly protrude along the $x$ and $y$ directions. Such errors can be reduced by using an unstructured mesh (Fig. 7e), and by calculating on a finer grid (Fig. 7f). By doing so, the calculated contours become closer to the expected circular pattern.

By performing tests on progressively finer meshes, we can also check the convergence of our CVFEM algorithm. For this purpose, we consider two predictive measures to assess grid convergence. The first is the normalized modeling error $(h-H)/h$ between the calculated deposit height $H$ and the analytical value $h = 0.241$ m. Noting that even unstructured meshes can introduce some bias (in particular when the mesh is coarse), our second measure is the difference between the maximum and minimum radii associated with the contour $\tilde{z} = 0.1h$, normalized by the analytical value $r_{10} = 1.628$ m.

In Table 1, we list these height and radius measures for different mesh sizes, as characterized by the average length of element edges and by the number of elements of the mesh. As the mesh is refined, we see that both measures converge to 0. In particular, the normalized modeling error $(h-H)/h$ clearly converges to first order with respect to element size. In Table 1, we also report the computational time in seconds needed to run these simulations on an i5-9500 Intel processor. We emphasize that the use of the dynamic time step in our solution contributes significantly to its efficiency. Preliminary versions of the code used a constant time step selected by $\Delta t = 0.25 \Delta \ell^2 / \nu^*$. This approach produces the same predictions as those reported here but requires over an order of magnitude more CPU time.

## 8   Comparisons with field and laboratory data

To further test the model, in the section we present comparisons with field and laboratory data. Measured profiles for the toes of debris flow deposits are first exploited, to verify the applicability of the critical slope and Mohr–Coulomb model to field cases. Comparisons with new laboratory experiments are then made, to check the ability of the CVFEM model to predict the overall morphology of cohesive-frictional deposits. The calibration and CVFEM numerical model code and input/output data discussed in this section are available in Chen et al. (2022).

## 8.1 Comparison with field profiles

Coussot et al. (1996) observed six natural debris flow deposits in the French Alps. By categorizing these deposits by their fines fraction (ratio of particles whose diameter is less than 40 $\mu$m to total solid volume), they found that debris flow deposits with a low fines fraction ($< 1\%$), at Bourgeat, Le Bez and Ste-Elisabeth, exhibit nearly straight profiles, whereas debris flow deposits with a high fines fraction (10%-15%), at Les Sables, St-Julien, and Mont Guillaume, exhibit significant snout-like toes. Coussot et al., therefore, focused on the latter case to test their model involving only the effect of yield stress. For each deposit with a high fines fraction, they documented two profiles, frontal and lateral, which they sought to fit by calibrating two parameters: the bed slope $\tan\beta$ and the yield stress over specific weight $\tau_Y/(\rho g)$. For each site, they calibrated these parameters separately for the frontal and lateral profiles. The lengths of the profiled deposits were in the range 2 to 15 m, and the corresponding thicknesses in the range 1.5 to 3 m.

Straight profiles, characterized by a constant slope, can be reproduced in our model by setting the yield stress to zero and the saturated friction slope $\tan\phi$ equal to the deposit surface slope. We therefore need to check whether our model can reproduce also the snout-like profiles observed for the case of high fines fraction. By taking both friction angle and yield stress into account, we can test whether analytical profiles can reproduce the field profiles using only one set of parameters per site. For this purpose, we assume that the frontal and lateral profiles at the same site share the same material properties ($\tan\phi$ and $\tau_Y/(\rho g)$). For the frontal profile, we treat the substrate bed slope ($\tan\beta$) as unknown, while for the lateral profile we assume that the bed slope is zero ($\tan\beta = 0$).

To estimate the three parameters, we fit the analytical solution given by Eq. (24) to the two measured profiles. Assigning the measured toe position as the boundary condition ($x_0 = x_{\text{toe}}$ and $H(x_0) = 0$), we obtain a predicted profile for given values of the frontal substrate bed slope $\tan\beta$, the saturated friction angle $\tan\phi$, and the ratio of yield stress over specific weight $\tau_Y/(\rho g)$. Then, on minimizing the root mean square error (RMSE) between predicted and measured fan profiles we arrived at best-fit estimates for $\tan\beta$, $\tan\phi$ and $\tau_Y/(\rho g)$.

In Fig. 8, we compare the resulting profiles with the field data, normalizing both axes by the length scale $\tau_Y/(\rho g)$. From the figure, we see that our critical slope model based on the Mohr–Coulomb constitutive law provides close fits to the field observations in the cases of Les Sables (Fig. 8a,b) and Mont Guillaume (Fig. 8e,f) and an acceptable fit in the case of St-Julien (Fig. 8c,d). The ability to use the same parameters (friction angle and yield stress) to fit both frontal and lateral profiles indicates that, while it is significant during the flowing stage, inertia may only play a limited role in determining the final deposit morphologies. For the debris flow deposits in Les Sables and St-Julien, the additional parameter ($\tan\phi$) plays an important role in determining the deposit morphology, and provides the degree of freedom needed to describe each pair of profiles for the same site using the same set of parameters. For Mont Guillaume, calibration produces a low value for the saturated friction angle, indicating that the yield stress and bed slope are sufficient to represent the deposit morphology. This may be due to the high clay content at this site.

Depending on scale and material composition, either the friction angle or the yield stress alone may be sufficient to characterize certain debris deposits in the field. Both influences, however, must be considered for intermediate cases, and to encompass

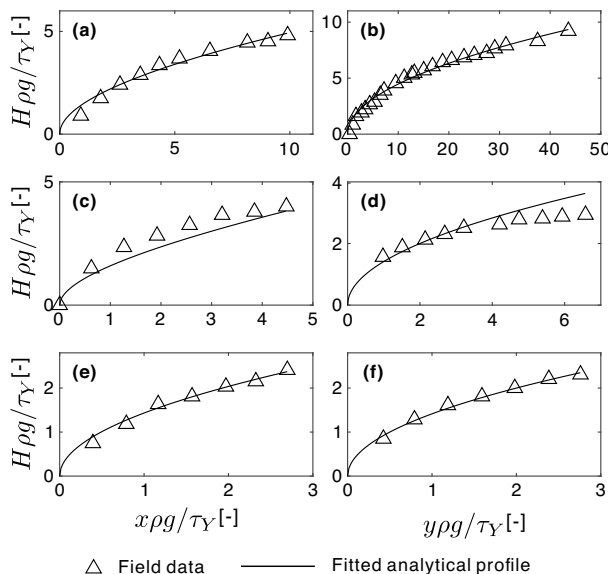

**Figure 8.** Comparison of debris deposit profiles at three field sites in the French Alps (Coussot et al., 1996) with analytical profiles calculated using calibrated values for parameters $\tan\beta$, $\tan\phi$ and $\tau_Y/(\rho g)$: (a,c,e) Frontal profiles; (b,d,f) lateral profiles; (a,b) Les Sables ($\tan\beta = 0.136$, $\tan\phi = 0.069$, $\tau_Y/(\rho g) = 0.297$ m); (c,d) profiles for St-Julien ($\tan\beta = 0.296$, $\tan\phi = 0.262$, $\tau_Y/(\rho g) = 0.432$ m); (e,d) profiles for Mont Guillaume ($\tan\beta = 0.245$, $\tan\phi = 0.028$, $\tau_Y/(\rho g) = 0.656$ m).

the range of possible behaviors in a single description. However, other effects could also cause or contribute to the formation of steep snouts in debris flows. For example, pore pressure loss at the front and margins (e.g. Iverson, 1997; Iverson and Vallance, 2001; Savage and Iverson, 2003; Iverson et al., 2010; Gray, 2018), and the frictional hysteresis of the angular sand particles (e.g. Félix and Thomas, 2004; Mangeney et al., 2007; Edwards et al., 2017; Rocha et al., 2019; Edwards et al., 2019). These effects are not currently included in our model.

To go beyond transect comparisons, in the next section we will use laboratory experiments to test the ability of our CVFEM model to simulate the complete morphology of cohesive-frictional deposits.

### 8.2 Experimental design and conditions

To investigate the morphology of cohesive-frictional deposits in well-controlled conditions, but more complex geometries, we conducted new laboratory experiments at the Hydrotech Research Institute of National Taiwan University. As illustrated in Fig. 9, these experiments were conducted in faceted flumes, assembled from bevelled wood panels. Different from alluvial fan experiments (Le Hooke and Rohrer, 1979; Whipple et al., 1998; Delorme et al., 2018; Savi et al., 2020), involving water and cohesionless sediment, here the deposits are built from mixtures of sand, kaolinite and water, mixed together thoroughly to behave as a cohesive-frictional material. To produce varied deposits, controlled volumes of these mixtures were supplied upstream of steep V-shaped canyons, and conveyed by these canyons to zones of milder topography where they could spread,

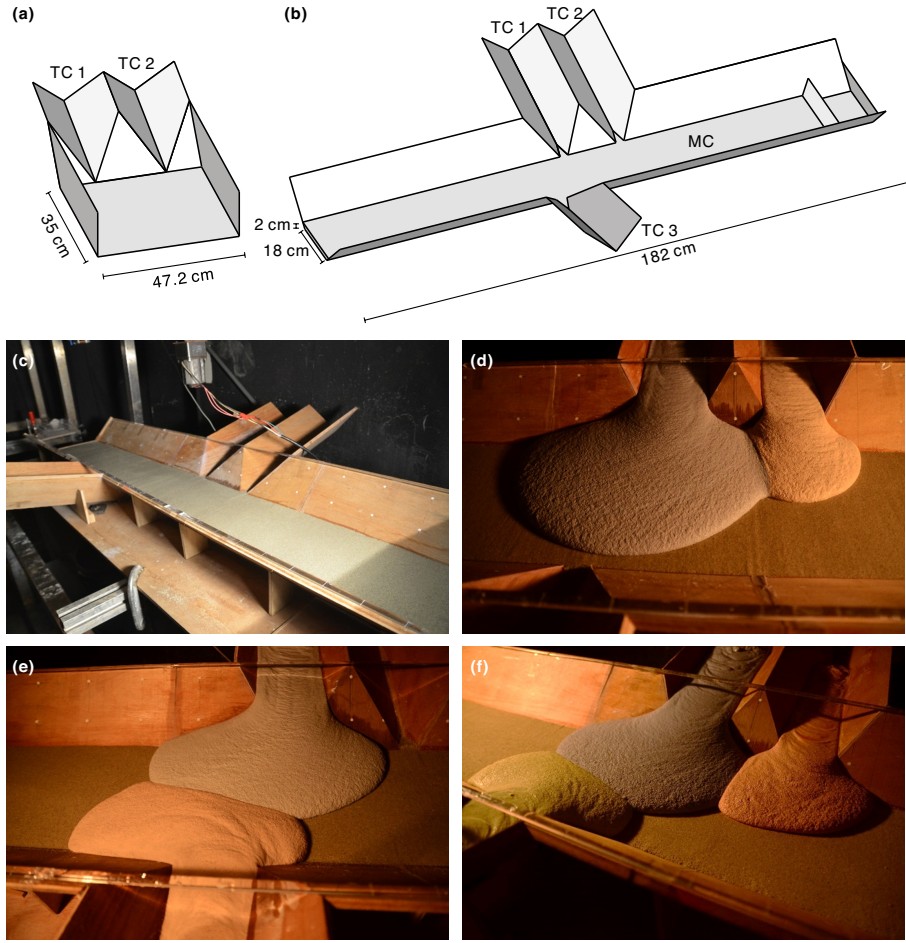

**Figure 9.** Experimental set-up and photos: (a) flume geometry for the canyon-plain experiments (T01–T04); (b) flume geometry for the canyon-valley experiments (T11–T15); (c) initial condition for runs T11–T14; (d,e,f) final deposits for runs T12, T13, and T14.

slow, and freeze in place. Water-soluble dyes were added to distinguish the materials supplied to different canyons. Finally laser scanning (Ni and Capart, 2006; Lobkovsky et al., 2007) was used to acquire high-resolution maps of the substrate and deposit topography.

As illustrated by the photographs of Fig. 9d–f, the experiments generate rather idealized deposits, which nevertheless reproduce various features exhibited by debris flow deposits in the field. These include steep snouts along lobe margins, and cusped

weld lines where separate lobes come into contact. Surface folds, indicative of viscoplastic behavior, can be observed at various locations (see for instance the lobe in the foreground of Fig. 9e), similar to the folds visible in some areas of the field deposit shown in Fig. 1.

Two series of tests were conducted: canyon-plain experiments (T01–T04), using the geometry shown in Fig. 9a, and canyon-valley experiments (T11–T15), using the geometry shown in Fig. 9b. For the canyon-plain experiments (runs T01–T04), two

V-shaped canyons connect to a wide U-shaped plain that has a planar floor and vertical walls. The canyon thalwegs have an inclination of 18.8 degrees relative to the planar floors. The experiments were designed so that the whole flume could be tilted away from horizontal, in the longitudinal direction of the tributary channels. In each run, a mixture of 61.4 wt% silica sand ($d_{50} = 0.6$ mm), 8.8 wt% kaolin, and 29.8 wt% water was used to deposit a fan into an initially empty and clean flume.

For run T01 the flume floor was horizontal, and two equal volumes of mixture were poured simultaneously upstream of the two canyons. For run T02 the inclination was the same, but the volumes supplied to the two tributary channels TC1 and TC2 were in a ratio of 1 to 2. The continuous mass input was arranged to start and end at the same time. Runs T03 and T04 were identical to run T02 apart from different flume tilt angles, set respectively to 3 and 6 degrees. For these runs, the topography was scanned with the laser oriented perpendicular to the canyons, and the resulting DEM data have resolution 2 mm x 2 mm.

For the canyon-valley experiments (T11–T15), the flume had a more complex configuration, illustrated in Fig. 9b. Three V-shaped canyons, having thalweg inclinations equal to 14 degrees, connect at right angles to a wide trapezoidal channel of longitudinal inclination equal to 3 degrees. Two of the canyons (TC1 and TC2) connect on the right side, and one on the left (TC3), slightly downstream. In all runs the initial state of the canyon was clean wood, but the main channel was covered by a 2 cm thick layer unconsolidated silica sand ($d_{50} = 0.6$ mm). For run T11 a controlled volume of mixture was supplied to tributary TC1 only. For run T12 different volumes were supplied simultaneously to tributaries TC1 and TC2, and arranged to start and end at the same times. For run T13 different volumes were supplied to tributaries TC1 and TC3, and for run T14 different volumes were supplied simultaneously to all 3 tributaries.

For run T15, deposits were formed in three separate stages. In the first stage, deposits were formed as in run T13 by supplying different volumes to tributaries TC1 and TC3. In the second stage, a constant water discharge was supplied to the main channel for 20 minutes, eroding the first stage deposits. The resulting topography was scanned to provide initial conditions for the third stage, in which new volumes of material were supplied to tributaries TC1 and TC3. This provides an opportunity to examine the formation of fresh deposits onto a pre-existing deposit surface. For all canyon-valley experiments, the topography was scanned with the laser oriented orthogonal to the main channel and parallel to the canyons, and the resulting DEM data have resolution 5 mm by 5 mm.

In Table , we present the range of parameter values covered by the laboratory experiments (runs T11-T15), and compare them to typical values for natural debris flows (Iverson, 1997; Zhou and Ng, 2010). From the table, we can see that the experiments exhibit smaller Froude and Reynolds numbers, hence inertia effects are smaller in the experiments than in field cases. Nevertheless, the Bagnold number (ratio between collisional and viscous forces), the Savage number (ratio between collisional and frictional forces) and Friction number (ratio between frictional and viscous forces) in the experiments are within the range of values encountered for natural debris flows.

In the next sections, the data from these different experiments will be used to calibrate model parameters, and compare CVFEM simulation results with the topography measurements acquired in each case.

**Table 2.** Parameter ranges in the laboratory experiments (T11-T15) and in documented field cases.

| Parameter | Symbol | Unit | Definition | Range in T11-T15 | Range in field cases* |
|---|---|---|---|---|---|
| Volumetric solids fraction in mixture | $\nu_s$ | - | | 0.412 | 0.3-0.72 |
| Volumetric fines fraction in interstitial fluid | $\nu_{fines}$ | - | | 0.100 | 0.02-0.12 |
| Solid density (silica sand and Kaolin) | $\rho_s$ | kg/m$^3$ | | 2650 | 2500-3000 |
| Interstitial fluid (Kaolin + water) density | $\rho_f$ | kg/m$^3$ | $\rho_s \nu_{fines} + \rho_w(1-\nu_{fines})$ | 1160 | 1030-1200 |
| Characteristic grain size | $\delta$ | m | $\delta \approx d_{50}$ | 0.0006 | 0.001-0.005 |
| Mean deposit thickness | $H$ | m | | 0.01-0.02 | 1-20 |
| Average front velocity | $u$ | m/s | | 0.015-0.02 | 10-20 |
| Flow front shear rate | $\dot{\gamma}$ | 1/s | $u/H$ | 0.75-2 | 1-20 |
| Interstitial fluid viscosity | $\mu$ | Pa·s | | 0.015-0.46** | 0.001-0.5 |
| Froude number | $Fr$ | - | $u/\sqrt{gH}$ | 0.034-0.064 | 1.4-3.2 |
| Reynolds number | $N_{Rey}$ | - | $\rho u H/\mu$ | 0.5-47 | $10^3$- $10^8$ |
| Bagnold number | $N_{Bag}$ | - | $\nu_s \rho_s \delta^2 \dot{\gamma}/((1-\nu_s)\mu)$ | 0.0011-0.1 | 0.002-20 |
| Savage number | $N_{Sav}$ | - | $\dot{\gamma}^2 \rho_s \delta^2/((\rho_s - \rho_f)(gH\tan\phi_s))$ | $3\times10^{-6}$-$4\times10^{-5}$ | $1\times10^{-7}$-$5\times10^{-2}$ |
| Friction number | $N_{Fric}$ | - | $N_{Bag}/N_{Sav}$ | $2\times10^2$-$3\times10^4$ | $1\times10^0$-$4\times10^5$ |

* The parameter ranges in documented field cases are collected or calculated from the data of Iverson (1997) and Zhou and Ng (2010).

** Viscosity for the experimental interstitial fluid is estimated from a set of viscometer measurements.

## 8.3 Comparison with canyon-plain experiments

To apply the CVFEM method to the canyon-plain experiments (runs T01–T04), we first determine model parameters from longitudinal deposit profiles, measured along the centrelines of the deposits from each canyon (see example profile locations in Fig. 10a). The calibration method used is the same as the one applied to the field profiles, except that the substrate slope $\tan\beta$ is known from the flume geometry, hence only the material parameters $\tan\phi$ and $\tau_Y/(\rho g)$ remain to be determined. For this set of experiments, some variability in material properties was caused by uncontrolled variations in moisture in the kaolin. For this reason, we use all eight of the available measured profiles together, to estimate a pair of parameters that best fit the whole series of experiments. The resulting estimates are $\tan\phi = 0.063$ and $\tau_Y/(\rho g) = 0.115$ cm.

Initial and boundary conditions are set up as follows. An unstructured mesh of average element size $\Delta\ell = 4$ mm is generated over the problem domain. The flume topography measured before each experiment is then used to set the substrate and initial surface elevations $z_b(x,y)$ and $\tilde{z}(x,y,0)$. To input the deposits, constant discharge sources are placed at the vertices closest to the upstream ends of the two channel thalwegs $(x,y) = (0,10)$ cm and $(x,y) = (0,34.8)$ cm, respectively. The rates of these discharges are set to ensure that, at the end of the chosen simulation time, the volumes supplied match the measured experimental volumes for each source.

In Fig. 10, we compare simulation results with the experimental measurements for the four runs T01–T04. Qualitatively and quantitatively, the simulations are found to predict reasonably well the measured topography of the deposits. As indicated by

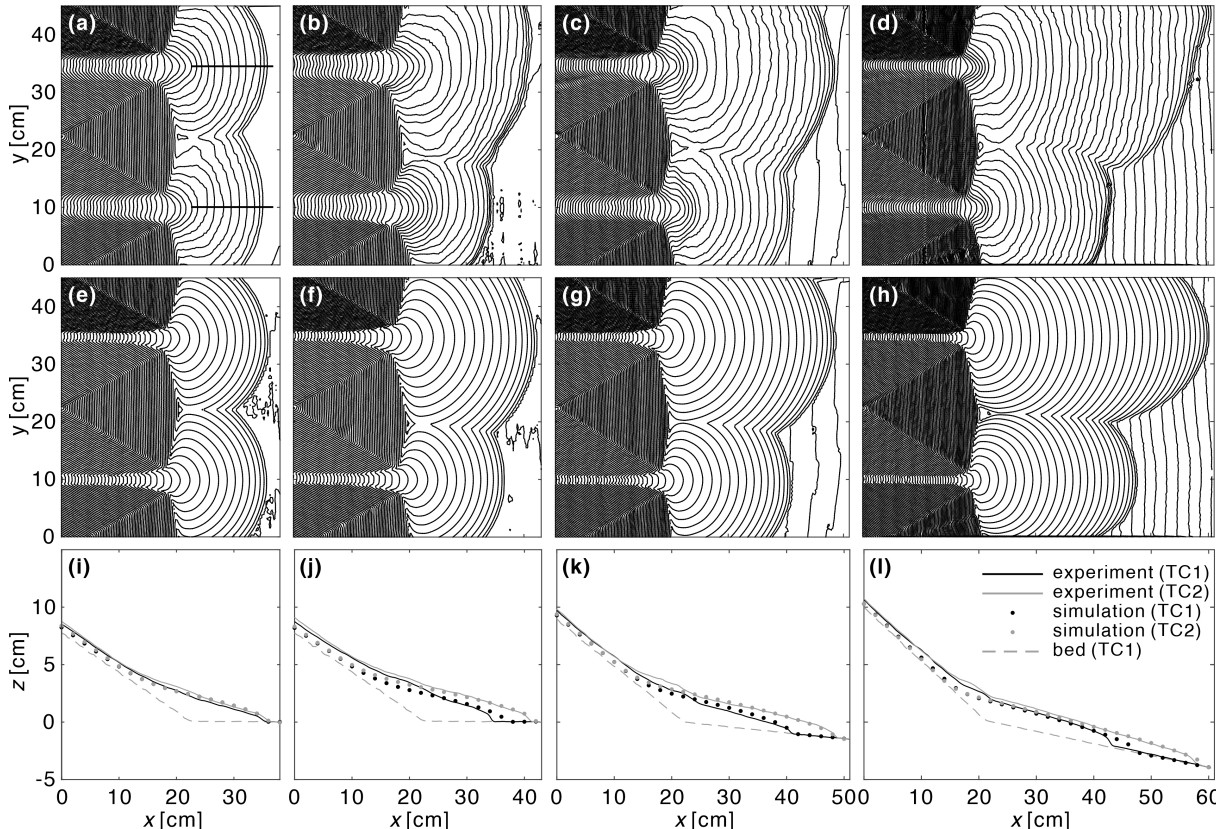

**Figure 10.** Comparison of measured and simulated deposit topographies for the canyon-plain experiments. Left to right: runs T01, T02, T03, and T04 corresponding to flume inclinations of 0, 0, 3, and 6 degrees; (a-d) experimental results; (e-h) CVFEM simulations; (i-l) longitudinal profiles for transects y = 10 cm (black) and y = 34.8 cm (gray), along the centerlines of the deposits. Contours at intervals $\Delta z = 0.2$ cm.

the contours, both the simulations and experiments produce steep snouts along lobe margins, well-defined cusps along weld lines, where two lobes come into contact, and saddle points along these same weld lines.

In planform (Fig. 10a–h), the model is able to reproduce well the outer boundaries of the deposits, both along the steep canyon and valley sides, and over the mildly inclined floor. Agreement holds for both the symmetric (equal volumes supplied to the two canyons) and asymmetric cases (unequal volumes). The model also reproduces the gradual elongation of the deposit lobes as the flume inclination is increased.

In profile (Fig. 10i–l), model results also compare well with the measurements. The model is able to capture the observed deposit slope variations, from steep upstream of the canyons, to mild over the thick lobes, back to steep snouts at the downstream toes. In both the simulations and experiments, furthermore, the deposits become gradually shallower as the flume slope is increased.

Nevertheless, there are some discrepancies between the CVFEM model and the experiments. Within the canyons and at canyon outlets, the model produces narrower and shallower deposits than the experimental results. This could be due to the

geometrical simplifications used to derive the critical slope model, in which the basal substrate was assumed approximately parallel to the surface. There are also some mismatches in planform length and width, possibly due to the previously mentioned moisture variations between runs. This is especially notable for the distal parts of run T04.

### 8.4 Comparison with canyon-valley experiments

For the canyon-valley experiments (T11–T15), the moisture was better controlled, hence the material composition was more nearly identical for all runs. We can therefore use the longitudinal profile for the single deposit produced in run T11 (red line in Fig. 11a) to calibrate the parameters for all cases. The resulting estimates for the material parameters, $\tan\phi = 0.118$ and $\tau_Y/(\rho g) = 0.344$ cm, are used for all CVFEM simulations of this series.

To simulate these runs, we use an unstructured mesh of average element size $\Delta\ell = 5$ mm.Like before, for each case we obtain the initial condition by sampling the measured pre-event topography at the mesh nodes. For runs T11–T14, we prescribe point sources of constant discharge at the vertices where canyon thalwegs intersect the domain boundaries (red points in Fig. 11). For run T15, the deposits partly buried the canyons, hence line sources are used instead at cross sections along the domain boundary (red lines in Fig. 11j). The discharge for these various sources are again set to match the volumes of the individual deposits.

To compare measured and simulated results, topographic contours and deposit thickness maps for the different cases are presented in Fig. 11. Overall, good agreement is observed between the CVFEM simulations and the experiments. Because the main channel dips to the left, the deposit lobes acquire an asymmetric, distorted shape, which is well-reproduced by the simulations. In both the experiments and the simulations, steep snouts are produced along the outer and side margins of the deposits, where they connect with the valley bed and sides. For runs T12 to T14, the weld lines obtained where different lobes come into contact are also accurately predicted. Using a single set of material parameters, the simulations also reproduce well the deposit thickness distributions obtained in the different experiments.

Similar to the canyon-plain experiments, some discrepancies are nevertheless observed between the simulations and experiments. The simulated fans are slightly wider ($x$ direction) and shorter ($y$ direction) than their experimental counterparts. This could be due to momentum, neglected in our CVFEM model, allowing the experimental mixture to flow out further in the canyon direction.

Finally, the T15 experiment (Fig. 11i,j) allows us to test our model for the case of fresh deposits onto a pre-existing deposit of complex shape. This case is similar to the 2009 Xinfa debris flow shown in Fig. 1, where the inundation of houses suggests a flow of around 2-3m deep occurring on a much larger (around 40m tall) pre-existing debris-flow fan. We see that the experimental deposit exhibits similar features to the Xinfa debris flow deposits, in particular the well defined snouts of the secondary deposits on top of the pre-existing deposit. In the experiments, the earlier deposit may deform slightly due to the new deposition, but we neglect this complication and take it as a new rigid boundary in the simulations. For this challenging case, the CVFEM model again provides an excellent overall prediction of the thickness, extent, and morphology of the secondary deposits. In both experiment and simulation, the fresh deposits do not completely cover the pre-existing lobes. The

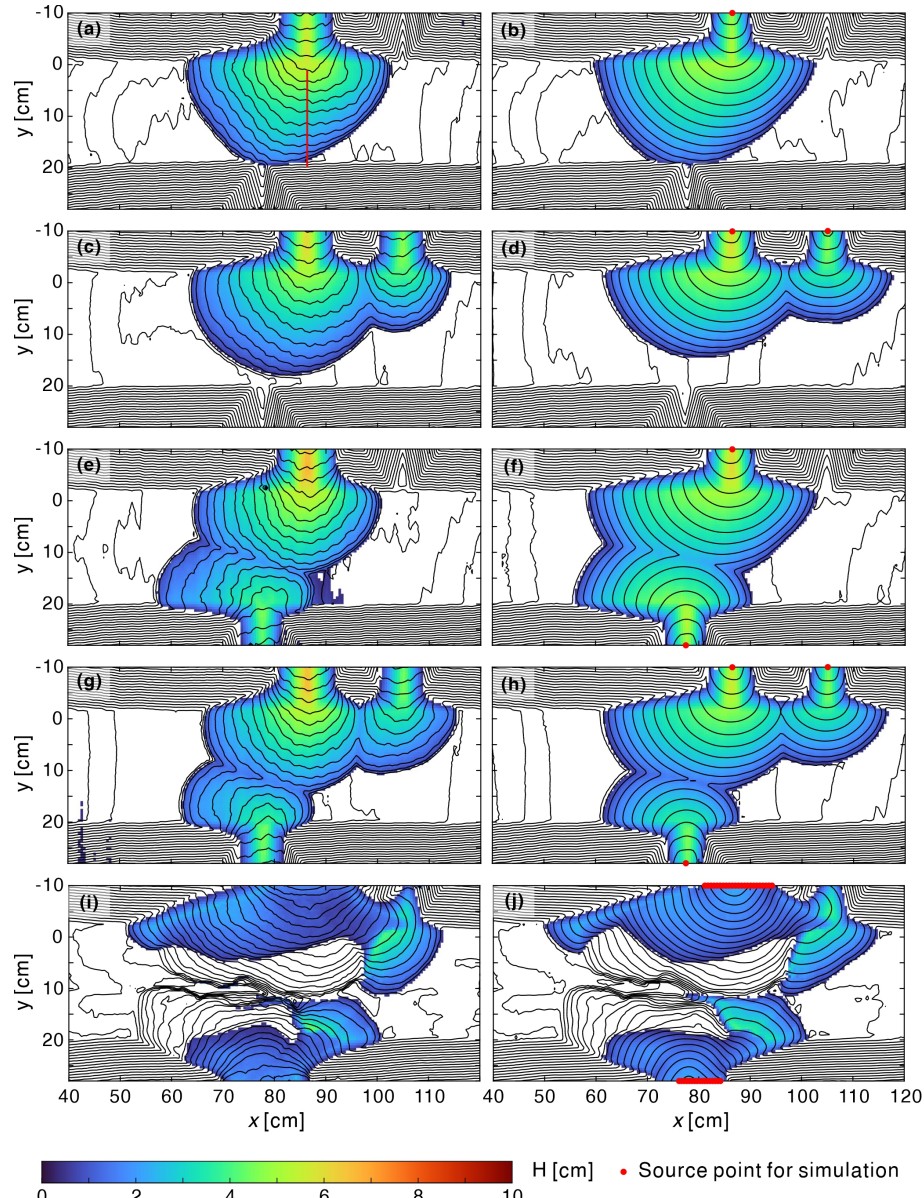

**Figure 11.** Comparison of measured (left) and simulated deposit topographies (right) for the canyon-valley experiments: (a,b) run T11; (c,d) run T12; (e,f) run T13; (g,h) run T14; (i,j) run T15. Lines: elevation contours at intervals $\Delta z = 0.5$ cm.

fresh material stops over these lobes at some locations, flowing further at other locations to form new secondary lobes. The corresponding margins again feature well-defined snouts.

## 9 Conclusions

In this paper, we proposed a novel computational model to simulate the morphology of debris flow deposits. The numerical algorithm uses the control volume finite element method (CVFEM) to discretely approximate fluxes over a finite element mesh, and explicitly enforce mass balance over prescribed control volumes. Unlike fluvial and mud flow deposits, debris flow deposits are affected by both cohesion and friction. To set the critical slope at which flow starts or stops, we therefore adopted a Mohr–Coulomb criterion that includes both a yield stress and a friction angle.

We verified the CVFEM algorithm by comparing computational results to analytical solutions in idealized cases, obtaining excellent agreement. Comparisons with field profiles were then performed to check that our critical slope model based on the Mohr–Coulomb relation can reproduce the key features of debris flow deposits. For deposits characterized by a high fines fraction, the inclusion of a yield stress allows our model to reproduce the blunted snouts observed at deposit toes. Accounting for a friction angle, on the other hand, allows our model to match the trailing slope observed away from the toes, and makes the model applicable also to deposits with a low fines fraction, which feature more even slopes.

Finally, comparisons with new laboratory experiments were conducted to test the ability of our CVFEM model to predict the extent, thickness and morphology of cohesive-frictional deposits in more complex geometries. The conditions considered include supply by single and multiple sources, and deposition over faceted substrates and pre-existing deposits. Using material parameters calibrated from one or more transects, the model is found to reproduce well the measured topography in all cases. Deposits features captured accurately by the model include steep snouts along the margins of primary and secondary lobes, and cusped weld lines where different lobes come into contact.

Although good agreement was obtained for the different comparisons, we do recognize some possible limitations. First, the model cannot simulate the dynamic evolution of debris flows, since it is only designed for computing the quasi-static morphology of debris flow deposits, and relies on a hypothetical diffusivity and pseudo time steps. Second, the model neglects flow momentum and basal erosion, hence it does not apply to rapid or erosive debris flows (Armanini et al., 2005). Besides, as noted above, the model does not include other effects that may lead to the formations of snouts and channel levees, such as pore pressure loss and frictional hysteresis. Likewise, it does not account for the thixotropic behavior whereby deposits gradually solidify to form a new substrate for fresh deposits (Murata, 1984; Roussel, 2006). Finally, our model and experiments do not include processes like channel formation, migration and avulsion that also affect the evolution over time of debris and alluvial fans (Le Hooke and Rohrer, 1979; Whipple et al., 1998; Delorme et al., 2018; Savi et al., 2020).

Despite these current limitations, we have shown that a critical slope model accounting for yield stress and friction angle can simulate deposit morphology with excellent efficiency using dynamic time steps. Aside from computation time, another key consideration is the work involved in calibrating model parameters. In this regard, an important advantage of our proposed simple model is that its parameters can be calibrated directly from topography profile data. As done in the paper for the experimental cases, all model parameters can be acquired from a single long profile through observed deposits. It is therefore not necessary to run the three-dimensional model multiple times to adjust model parameters by trial and error. More complex models, by contrast, typically require multiple iterations, or must rely on other sampling and material analysis to acquire

parameter values. The model can easily calibrated parameters for a broader range of conditions than considered previously.
To simulate such deposits in complex geometries, moreover, the control volume finite element method (CVFEM) was found to provide a promising numerical approach, and could possibly be extended in the future to more general processes or other geomorphic systems.

*Code and data availability.* The Matlab codes of the CVFEM model and parameter calibrations, input data (experimental pre-event and post-event topography data), and model output data are available at Chen et al. (2022), https://doi.org/10.5281/zenodo.7324739.
The algorithm for constructing unstructured mesh used in this paper is an open access Matlab package built by Engwirda (Engwirda, 2014) available at https://github.com/dengwirda/mesh2d.

The algorithms for linearly interpolating triangulation and plotting contours for triangular mesh used in this paper are open access Matlab codes built by Hanselman (Hanselman, 2021a, b) available at https://www.mathworks.com/matlabcentral/fileexchange/38925 and https://www.mathworks.com/matlabcentral/fileexchange/38858.

*Author contributions.* T-YKC performed numerical simulations, contributed to designing the research methodology, analyzed data, created figures, and wrote the first draft; Y-CW conducted laboratory experiment, contributed to designing the research methodology, and analyzed data; C-YH discussed the results, contributed to the writing, visualization, and funding acquisition; HC contributed to designing the research methodology, experiment design and setup, discussed the results, project advising, funding acquisition, and contributed to the visualization, writing and editing of the manuscript; VRV contributed to designing the research methodology, discussed the results, project advising, 505 resources, and contributed to the visualization, writing and editing of the manuscript.

*Competing interests.* The authors declare no competing interests relevant to this study.

*Acknowledgements.* Useful feedback regarding the implementation and applications of our proposed scheme was provided by Philippe Delandmeter (Fugro Engineers Belgium). The research was supported by: the Young Scholar Fellowship Program by Ministry of Science and Technology (MOST) in Taiwan, under Grant MOST110-2636-M-005-001; the Dragon Gate Program by MOST in Taiwan, under Grant 510 MOST108-2911-I-002-564.

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
