# Peer review of "A control volume finite element model for predicting the morphology of cohesive-frictional debris flow deposits"

_Earth Surface Dynamics, 2022_

## Author Comment (AC1)

**Response to reviewer comment 1 (RC1) by Chris Johnson**

Tzu-Yin Kasha Chen1, Ying-Chen Wu1, Chi-Yao Hung2, Hervé Capart1, and Vaughan R. Voller3 1Dept of Civil Engineering and Hydrotech Research Institute, National Taiwan University, Taiwan 2Dept of Soil and Water Conservation, National Chung-Hsing University, Taiwan 3Department of Civil, Environmental, and Geo- Engineering, University of Minnesota, USA

Correspondence: Tzu-Yin Kasha Chen (d06521004@ntu.edu.tw)

We wish to thank Dr. Chris Johnson for his constructive comments. We reproduce below the review comments in italics, followed by our reply to each comment in normal type. The possible corresponding changes made to the manuscript are reproduced in blue.

**5 Referee: 1**

**Review Comment:**

This well-written paper combines theory, computational methods, analogue laboratory experiments and field observations of cohesive debris flows.

The paper presents a depth-integrated model for cohesive-frictional debris flows. The model resembles lubrication theory,

10 in that inertia of the fluid is neglected, and the depth-integrated flux is therefore an instantaneous function of the local surface gradient and thickness.

These model equations are solved using a control volume finite-element technique, which is validated by comparison to exact solutions. The model predicts very well the deposits of laboratory experiments of sand/kaolinite/water flows, and is compared with with field observations from Coussot et al. (1996).

15 The agreement between the model predictions and laboratory experiments is striking, and I have no doubt that the modelling approximations made (shallow inertia-free flow with homogeneous cohesive-frictional rheology) are very well suited to the flows in the lab. As a description of laboratory experiments it is therefore a strong piece of work.

**Our reply:**

20 We thank Dr. Johnson for appreciating the manuscript, and describe below the changes made to address the major and minor review comments.

**Review Comment:**

25 However, it is much less clear to me that the physics studied here is relevant to many natural debris flows. There is some acknowledgement of the differences between the modelling here and natural debris flows (lines 52-58 and 441-446). But in my

view there should be a much more comprehensive discussion and justification of which predictions from this paper (obtained at laboratory scale) would be expected to hold true at field scale, and which would not. For example:

- 1. It is not clear that yield stress is responsible for the blunt snouts of natural debris flows. In the model used by this paper,
- 30 the yield stress required to produce blunt snouts at field scale is very large, evidenced by a fit of  $\tau Y/(\rho g) \approx 0.5m$  in the field observations, compared to 0.001m in the experiments. How can this difference of a factor of 500 in yield stress be explained?

It seems likely to me that the formation of blunt snouts at field scale could due to a different process, for example the loss of excess pore pressure at the flow front, resulting in a substantial increase in the frictional part of the stress here. That

35

is, the blunt snout could be due to a rheology that is inhomogeneous but not cohesive. Section 8.1 would benefit from some discussion of these points.

**Our reply:**

Thank you for the comment. We agree that the formation of blunt snouts could be due to different processes. We propose adding the following discussion in Section 8.1:

40 However, other effects could also cause or contribute to the formation of steep snouts in debris flows. For example, pore pressure loss at the front and margins (e.g. Iverson, 1997; Iverson and Vallance, 2001; Savage and Iverson, 2003; Iverson et al., 2010; Gray, 2018), and the frictional hysteresis of the angular sand particles (e.g. Félix and Thomas, 2004; Mangeney et al., 2007; Edwards et al., 2017; Rocha et al., 2019; Edwards et al., 2019). These effects are not currently included in our model.

**45 And we also propose clarifying this limitation of our model in the conclusion:**

First, the model cannot simulate the dynamic evolution of debris flows, since it is only designed for computing the quasi-static morphology of debris flow deposits, and relies on a hypothetical diffusivity and pseudo time steps. Second, the model neglects flow momentum and basal erosion, hence it does not apply to rapid or erosive debris flows (Armanini et al., 2005). Besides, as noted above, the model does not include other effects that may lead to the formations of snouts and channel levees, such as

50 the pore pressure loss and the frictional hysteresis. Likewise, it does not account for the thixotropic behavior whereby deposits gradually solidify to form a new substrate for fresh deposits (Murata, 1984; Roussel, 2006). Finally, our model and experiments do not include processes like channel formation, migration and avulsion that also affect the evolution over time of debris and alluvial fans (Le Hooke and Rohrer, 1979; Whipple et al., 1998; Delorme et al., 2018; Savi et al., 2020).

**55 *Review Comment:**

2 More generally, the difference in physics between small-scale analogue experiments and natural-scale flows has been raised by several authors, including Iverson (e.g. https://doi.org/10.1016/j.geomorph.2015.02.033). The paper would benefit from a discussion of the parameter regime realised in experiments (Froude number, Reynolds number, Savage/inertial number etc.) and a comparison of this with natural examples.

Table 2. Parameter ranges in the laboratory experiments (T11-T15) and in documented field cases.

| Parameter                                       | Symbol             | Unit              | Definition                                                              | Range in T11-T15                      | Range in field cases*                 |  |
|-------------------------------------------------|--------------------|-------------------|-------------------------------------------------------------------------|---------------------------------------|---------------------------------------|--|
| Volumetric solids fraction in mixture           | $v_s$              | -                 |                                                                         | 0.412                                 | 0.3-0.72                              |  |
| Volumetric fines fraction in interstitial fluid | v fines | -                 |                                                                         | 0.100                                 | 0.02-0.12                             |  |
| Solid density (silica sand and Kaolin)          | $ ho_s$            | kg/m 3 |                                                                         | 2650                                  | 2500-3000                             |  |
| Interstitial fluid (Kaolin + water) density     | $ ho_f$            | kg/m 3 | $\rho_s v_{fines} + \rho_w (1 - v_{fines})$                             | 1160                                  | 1030-1200                             |  |
| Characteristic grain size                       | δ                  | m                 | $\delta \approx d_{50}$                                                 | 0.0006                                | 0.001-0.005                           |  |
| Mean deposit thickness                          | Н                  | m                 |                                                                         | 0.01-0.02                             | 1-20                                  |  |
| Average front velocity                          | и                  | m/s               |                                                                         | 0.015-0.02                            | 10-20                                 |  |
| Flow front shear rate                           | Ý                  | 1/s               | u/H                                                                     | 0.75-2                                | 1-20                                  |  |
| Interstitial fluid viscosity                    | μ                  | Pa·s              |                                                                         | 0.015-0.46**                          | 0.001-0.5                             |  |
| Froude number                                   | Fr                 | -                 | $u/\sqrt{gH}$                                                           | 0.034-0.064                           | 1.4-3.2                               |  |
| Reynolds number                                 | N Rey   | -                 | $ ho u H/\mu$                                                           | 0.5-47                                | $10^3 - 10^8$                         |  |
| Bagnold number                                  | N Bag   | -                 | $v_s \rho_s \delta^2 \dot{\gamma} / ((1 - v_s)\mu)$                     | 0.0011-0.1                            | 0.002-20                              |  |
| Savage number                                   | N Sav   | -                 | $\dot{\gamma}^2 \rho_s \delta^2 / ((\rho_s - \rho_f) (gH \tan \phi_s))$ | $3 \times 10^{-6} - 4 \times 10^{-5}$ | $1 \times 10^{-7} - 5 \times 10^{-2}$ |  |
| Friction number                                 | $N_{Fric}$         | -                 | $N_{Bag}/N_{Sav}$                                                       | $2 \times 10^2 - 3 \times 10^4$       | $1 \times 10^{0} - 4 \times 10^{5}$   |  |

\* The parameter ranges in documented field cases are collected or calculated from the data of Iverson (1997) and Zhou and Ng (2010).

\*\* Viscosity for the experimental interstitial fluid is estimated from a set of viscometer measurements.

**60 Our reply:**

Thank you for this recommendation. We propose adding a table in Section 8.2 (see below) to present the range of parameters in the experiments and comparing to the values for field cases collected and calculated from the data of Iverson (1997) and Zhou and Ng (2010).

**65 We propose adding the following discussion in the text:**

In Table 2, we present the range of parameter values covered by the laboratory experiments (runs T11-T15), and compare them to typical values for natural debris flows (Iverson, 1997; Zhou and Ng, 2010). From the table, we can see that the experiments exhibit smaller Froude and Reynolds numbers, hence inertia effects are smaller in the experiments than in field cases. Nevertheless, the Bagnold number (ratio between collisional and viscous forces), the Savage number (ratio between collisional and

70 frictional forces) and Friction number (ratio between frictional and viscous forces) in the experiments are within the range of values encountered for natural debris flows.

**Review Comment:**

75 3 An important feature of the model is that the inertia of the flow is neglected (line 76). Is this really valid for natural debris flows? Many debris flows are supercritical and exhibit features such as shocks, roll waves and superelevation in curved channels, which require inertia. Is there evidence that inertia can be neglected at field scale?

**Our reply:**

The effects noted by Dr. Johnson are relevant in the flowing stage. However, our model focuses on the deposition stage of debris 80 flows, for which inertia is comparatively less important. The field data from Coussot et al. (1996) can serve as evidence. For two of the cases in this field data set, the modelled profiles, using the same pair of parameters, match the surveyed profiles well in two directions (longitudinal and transverse) and acceptably in the third case. If the deposit shape was strongly influenced by inertia, we would expect greater differences between longitudinal and transverse profiles.

We propose adding the following discussion in Section 8.1:

From the figure, we see that our critical slope model based on the Mohr-Coulomb constitutive law provides close fits to 85 the field observations in the cases of Les Sables (Fig. 8a,b) and Mont Guillaume (Fig. 8e,f) and an acceptable fit in the case of St-Julien (Fig. 8c,d). The ability to use the same parameters (friction angle and yield stress) to fit both frontal and lateral profiles indicates that, while it is significant during the flowing stage, inertia may only play a limited role in determining the final deposit morphologies.

90

**Review Comment:**

4 It is tempting to attribute the similar conical shapes of the natural debris flow fan (figure 1) and experimental deposits

95

(figure 9) to a similar formation mechanism. Though I do not know the 2009 Xinfa debris flow shown in figure 1, the inundation of houses in this figure suggests a flow of perhaps 2m deep occurring on a much larger (perhaps 30m tall?) pre-existing debris-flow fan. If so, this is clearly a completely different mechanism from the en masse deposition of the entire fan in the experiments. There is some acknowledgement of this around line 55, but in my view a much clearer statement is needed as to the differences between the modelling/experiments in this paper and the natural deposit in figure 1.

**100 Our reply:**

We clarify that only the last experimental case (run T15) features conditions similar to the field case of Fig. 1, in which new flows deposit on pre-existing fans.

To address this point, we propose to modify the text as follows:

Finally, the T15 experiment (Fig. 11i,j) allows us to test our model for the case of fresh deposits onto a pre-existing deposit of complex shape. This case is similar to the 2009 Xinfa debris flow shown in Fig. 1, where the inundation of houses suggests a flow 105 of around 2-3m deep occurring on a much larger (around 40m tall) pre-existing debris-flow fan. We see that the experimental deposit exhibits similar features to the Xinfa debris flow deposits, in particular the well defined snouts of the secondary deposits on top of the pre-existing deposit.

110

**Review Comment:**

Minor points:

*Equation (1):* There is no source term in this equation corresponding to the inflow. Is the inflow flux Qin modelled as a source term of limited spatial extent on the right hand side of (1)?

115

**Our reply:**

Indeed, the inflow flux  $Q_{in}$  is modelled as a source term of limited spatial extent on the right hand side of Eq. (1). To clarify this point, we propose to add this source term explicitly to Eq. (1), to be corrected to:

$$\frac{\partial \tilde{z}}{\partial t} = -\nabla \cdot \mathbf{q} + Q_{in} \delta(\mathbf{x}_{s}), \qquad (1)$$

120 where  $\delta$  is the Dirac delta function,  $\mathbf{x}_s = (x_s, y_s)$  is the location of the source, and  $Q_{in}$  is the inflow source volumetric flux. We propose to also explicitly add  $Q_{in}$  to the corresponding numerical statement, and correct Eq. (8) to:

$$\frac{\tilde{z}_{i}^{\text{new}} - \tilde{z}_{i}}{\Delta t} = -\frac{1}{A_{CV,i}} \sum_{j=1}^{m} Q_{j} + \frac{Q_{in,i}}{A_{CV,i}},$$
(8)

**125 Review Comment:**

*Equation (3) / line 83: I initially misunderstood the statement on line 83 and believed that Sc was a constant for a given material. It may be useful to make it clearer that Sc is dependent on local instantaneous flow depth and free-surface slope, and that this dependence is derived in section 4.*

**130 Our reply:**

We follow the advice and propose adding after Eq. (3):

and where  $S_c$  is a critical slope dependent on material properties and on the local instantaneous thickness of the flow layer. This dependence of  $S_c$  on the flow layer thickness is derived in section 4.

**135**

**Review Comment:**

**Equation (4):** how is this equation derived? From (3), it is clear that the free surface slope is no greater than  $S_c$ , but not clear to me why it is exactly equal to  $S_c$ . (Derivation of (4) must require some constraints on the initial conditions or inflow functions  $Q_{in}$ , as for certain choices of these are counterexamples to (4). For example, if zb(x, y) = 0 and  $Q_{in}(x, y) = k$  and

140 *the initial conditions are*  $\tilde{z} = 0$  *at* t = 0*, then the exact solution is*  $\tilde{z} = k \times t$ *, which does not satisfy equation 4.) In a recent paper*

(https://doi.org/10.1017/jfm.2021.1074, section 7) we referred to regions of a dry granular flow deposit that do satisfy (4) as "maximal", but this was not true of the entire deposit.

Our reply:

- Our governing equations (Eqs. (1)-(4)) are proposed and derived only for the regions where materials deposit from flowing to stopping. On the contrary, regions where there are no horizontal fluxes throughout the process, such as the domain in the counterexample that Dr. Johnson raised, are not the targets of our equations. When describing materials deposition, the flux **q** in Eq. (2) decreases from a non-zero value to zero. To provide such changes in the flux, the local slope  $||\nabla \tilde{z}||$  in Eq. (3) must first exceeds the critical slope  $S_c$  and finally decreases to exactly the critical slope  $S_c$  (3). When the local slope  $||\nabla \tilde{z}||$  equals
- the critical slope  $S_c$ , there is no flux, and therefore there is no way to further decrease the local slope to a value smaller than the critical slope. We recognize that Eq. (4) may not be valid when inertia effects are dominant; here, as is observed in the dry granular flows noted by Dr. Johnson, it may be possible to have outfluxing at values below the critical slope. To clarify the derivation of Eq. (4), we propose to modify the text as follows:

With this model, the flux is only non-zero when the local slope  $||\nabla \tilde{z}||$  exceeds the critical slope  $S_c$ . By contrast, models that 155 consider momentum effects can produce local deposit slopes that are smaller than critical slopes (e.g. Tregaskis et al., 2022). In our model, on the deposit surface where the flow slows down from motions to a complete stop, the flux **q** vanishes as the local slope  $||\nabla \tilde{z}||$  decreases from a value that exceeds the critical slope  $S_c$  to exactly the critical slope  $S_c$ , imposing the mathematical condition that

 $||\nabla \tilde{z}|| = S_c$

(9)

160 everywhere on the final deposit surface. Make particular note that the critical slope developed in our model (see Section 4) will involve the sum of two components, a constant friction slope and a yield stress term that will be an inverse function of the deposits thickness, thus the final slope over the predicted deposited debris flow may not take a constant value.

**165 Review Comment:**

*Figure 5 / line 209:* What is the source function Qin for these solutions? Presumably the source is at a different value of x for each flux?

Our reply:

170 Yes, the sources are positioned at different locations x for these cases, so that the resulting deposits have the same toe location. We propose adding the followin explanation to the text:

For each case and deposit height *H*, we supply material at a single point corresponding to the apex of each deposit. To facilitate comparison, the source locations are adjusted so that the deposits have the same toe location. These locations  $x_s$  are determined using the formula  $x_s = (AH - B\ln(|AH + B|))/A^2 - C$ , where  $A = -\tan\beta + \tan\phi$ ,  $B = \tau_Y/(\rho g)$ ,  $C = (-B\ln(B))/A^2$ .

180

**Review Comment:**

*Line 235:* I don't fully understand "an approximate analytical solution obtained by setting tan  $\beta = 0$ ": why does tan  $\beta=0$  allow an analytical solution, and what exactly is being compared (is a numerical solution with  $\beta \neq 0$  compared to an exact solution with  $\beta = 0$ ?)

**Our reply:**

We agree the description was confusing. To clarify, we propose to modify the text as follows:

For the longitudinal profile (Fig. 6c,g,k), we can use the analytical solution in Eq. (24) as the exact solution. For the transverse profile (Fig. 6d,h,l), the transect is not a true symmetry axis; therefore, we can only use the analytical solution obtained by setting  $\tan \beta = 0$  as an approximated solution. For each case, we impose a fixed thickness of the deposit at the origin for both analytical solutions and numerical solutions.

**190 Review Comment:**

Figure 6: Should "(b,f,j) transverse deposit profiles" read "(d,h,l) transverse deposit profiles"?

**Our reply:**

Yes, thank you for flagging this error in the caption. We will correct the caption as "(d,h,l) transverse deposit profiles". 195

**Review Comment:**

**Table 1:** What is the order of convergence of the numerical scheme in space and time? From the time discretisation in equation(8), it appears to be first order in time. Is it also first order in space?

**200**

**Our reply:**

The discretization (Eq. (8)) of the governing equations (Eq. (1)) is first order in time and second order in space. For the convergence of the numerical scheme, we don't look at space and time separately because the time steps are decided by element sizes. In the previous version, we use constant time steps  $\Delta t = 0.25\Delta \ell^2 / v^*$ . In the current version, we adopt dynamic time steps to improve the model efficiency. Please see the updated Table 1 below:

205

To clarify this point, we propose to add the following description to the text:

To speed up computations yet insure numerical stability, we use a dynamic time step  $\Delta t = 0.2\Delta \ell^2 / (v^* \max((\nabla \tilde{z}_{ele} - Sc_{ele}) / (\nabla \tilde{z}_{ele}))))$

| Table | 1. | Influence | of mesh | ı size | on | model | accuracy | and | com | putational | time. |
|-------|----|-----------|---------|--------|----|-------|----------|-----|-----|------------|-------|
|       |    |           |         |        |    |       | -        |     |     |            |       |

| Avg. element size [m] | # of elements | (h-H)/h | $(R_{10\rm max} - R_{10\rm min})/r_{10}$ | Computational time [s] |
|-----------------------|---------------|---------|------------------------------------------|------------------------|
| 0.265                 | 526           | 0.063   | 0.092                                    | 0.092                  |
| 0.132                 | 2116          | 0.032   | 0.037                                    | 2.46                   |
| 0.066                 | 8612          | 0.020   | 0.012                                    | 46.5                   |
| 0.033                 | 33986         | 0.011   | 0.007                                    | 1117.4                 |

where subscript  $_{ele}$  represents values at elements, and  $\Delta \ell$  is the average element size. For time step where max $((\nabla \tilde{z}_{ele} - 210 \ Sc_{ele})/(\nabla \tilde{z}_{ele}) \le 0$ , we use  $\Delta t = 0.25 \Delta \ell^2 / \nu^*$ .

From the updated Table 1, we can see that the modeling errors on the deposit thickness, (h - H)/h, decrease linearly as the average element size is reduced.

**215 *Review Comment:**

*Figure 10 and 11:* these contour plots are noticeably slow to plot in my PDF viewer: are they particularly large figures that could be reduced in resolution?

**Our reply:**

220 To address this problem, the figure format for Figures 10 and 11 will be changed from .PDF format to .JPG format (as shown below).

---

## Author Comment (AC2)

**Response to reviewer comment 2 (RC2) by Stefan Hergarten**

Tzu-Yin Kasha Chen[1], Ying-Chen Wu[1], Chi-Yao Hung[2], Hervé Capart[1], and Vaughan R. Voller[3]

[1]Dept of Civil Engineering and Hydrotech Research Institute, National Taiwan University, Taiwan
[2]Dept of Soil and Water Conservation, National Chung-Hsing University, Taiwan
[3]Department of Civil, Environmental, and Geo- Engineering, University of Minnesota, USA

**Correspondence:** Tzu-Yin Kasha Chen (d06521004@ntu.edu.tw)

We wish to thank Professor Stefan Hergarten for his constructive comments. We reproduce below the review comments in italics, followed by our reply to each comment in normal type. The possible corresponding changes made to the manuscript are reproduced in blue.

5 **Referee: 2**

*Review Comment:*

*In this paper, a theoretical and numerical model for the morphology of the deposits of debris flows is presented. As a main simplification compared to existing models, effects of inertia are neglected. While existing models are based on shallow-water type (Savage-Hutter) equations, this approach arrives at a nonlinear diffusion equation with a threshold slope. For validation,*

10 *analytical solutions, topographies of real debris flow deposits, and laboratory experiments (being a part of the study) are used.*

*First, I would like to emphasize that both the theory and the numerical implementation are described very well and in great detail. Since the diffusion equation is numerically not very challenging, one might even ask whether such a detailed and basic level is necessary. However, I do not complain about this.*

15 Our reply:

We thank Professor Hergarten for appreciating the manuscript, and describe below the changes made to address the major and minor comments.

20 *Review Comment:*

*My main criticism concerns the simplification by neglecting effects of inertia. As stated by the authors, this limits the applicability to low velocities. The question whether this is a serious limitation for the application to real debris flow is not addressed sufficiently. All results used for validation are solely based on the final final topography and thus on the very end of the movement when the velocities should indeed be small. On the other hand, the introduction starts from the hazard of debris flow, where*

25 *the runout length is more important than the morphology of the deposits. So the authors should point out more clearly that the referenced existing models also attempt to predict the runout also at high velocities, while this is not tested for the new model. It even looks as if the new model mainly constructs a final deposit topography that obeys a predefined relation between slope*

*and thickness.*

30   Our reply:

Thank you for raising the concerns. This is indeed what we do. To address this point, we propose to add the following clarification to the introduction:

Unlike existing models that also attempt to predict the runout at high velocities, we limit our scope and focus on predicting the final deposit morphologies of debris flows, modelled as slow, quasi-static processes.

35

*Review Comment:*

*As a second point concerning neglecting effects of friction, I am not fully convinced that it makes things simpler or more efficient. It is stated that the existing models require a large amount of input data. However, can go back to the original Savage-Hutter*

40   *equations with a simple static friction term and nothing else. Then the coefficient of friction would be the only model parameter. We could also go a step further and use the Mohr-Coulomb criterion as proposed in the recent manuscript. The number of parameters and their meaning would be almost the same in both models then. This scenario would allow for an assessment of how much we lose by neglecting effects of inertia and how much we save. Theoretically, we save much because the equations become simpler. However, the results about the computational performance given in Table 1 are disappointing. It seems that*

45   *the diffusion model model with the explicit time step requires very small time increments. Without having data for comparison available, it looks to me as if the new model was quite inefficient compared to existing models.*

*To summarize these points, it would be essential for me to see a thorough analysis of what we lose with regard to real debris flow with the new model and whether there is any increase in numerical efficiency.*

50   Our reply:

As pointed out by the Reviewer, the numerical model as described in the previous version of the manuscript was not computationally very efficient. To improve on this point, we have now modified our algorithm so that dynamic time steps can be used instead of constant time steps. By doing so, the model efficiency is now greatly improved, reducing the computational times by more than an order of magnitude (see updated Table 1 below). Further, we see a similar speedup in the simulations of

55   experiment cases.

Aside from computation time, another key consideration is the work involved in calibrating model parameters. In this regard, an important advantage of our proposed simple model is that its parameters can be calibrated directly from topography profile data. As done in the paper for the experimental cases, all model parameters can be acquired from a single long profile through

60   observed deposits. It is therefore not necessary to run the three-dimensional model multiple times to adjust model parameters by trial and error. More complex models, by contrast, typically require multiple iterations, or must rely on other sampling and material analysis to acquire parameter values.

**Table 1.** Influence of mesh size on model accuracy and computational time.

| Avg. element size [m] | # of elements | $(h - H)/h$ | $(R_{10\,max} - R_{10\,min})/r_{10}$ | Computational time [s] |
|---|---|---|---|---|
| 0.265 | 526 | 0.063 | 0.092 | 0.092 |
| 0.132 | 2116 | 0.032 | 0.037 | 2.46 |
| 0.066 | 8612 | 0.020 | 0.012 | 46.5 |
| 0.033 | 33986 | 0.011 | 0.007 | 1117.4 |

65

*Review Comment:*

*Provided that this can be done, I would also suggest to consider the following aspects:*

70  *Section 2: If I got it correctly, the flux is only dependent on the slope, but not on the thickness (above a minimum thickness). This means that the flow velocity increases with decreasing thickness. I would have rather expected a flow velocity that depends on the slope only. I guess that the rather high fluxes at low thickness arising from the approach used here are not very good for the numerical performance. Is there a specific reason for this approach?*

Our reply:

75  We thank Professor Hergarten for this interesting suggestion. Indeed when comparing the verification test runs, where the deposit thickness changes systematically from input to toe, modifying our diffusivity in the manner suggested, resulted in an additional 3 fold speed up in calculation time. However in simulating the experiments, where changes in the deposit thickness are not as systematic, there was a 3 fold slow down. So while the idea of exploring how modification of the diffusivity term may influence CPU is worthwhile, based on the results we have, we will retain our current algorithm, with the adjustable time

80  step. As noted above this provides a 10 fold speedup in calculation time from the algorithm used in the original submission.

*Review Comment:*

*Section 3: Rather for curiosity (since I am not an expert on this): Why did you not use a standard Delaunay triangulation in combination with Voronoi polygons as control volumes?*

85

Our reply:

Using Voronoi is an intriguing suggestion. In general, however, the vertices of the Voronoi cells do not coincide with the centers of the Delaunay triangles, and may even lie outside these triangles. By forcing the edges of the control volumes (CV) to have their vertices either at the centers of our triangular elements, or at the midpoints of the triangular sides, better accuracy can be

achieved. Moreover, equal weights can be used to interpolate vertex values from node values, which facilitates calculation and bookkeeping.

*Review Comment:*

*Equation 10: How did Qin come in here compared to Eq. 8, and what is it used for? I thought you start the simulation with a given thickness distribution. Or is it just the source term for reproducing the laboratory experiments?*

Our reply:

Indeed, the inflow flux $Q_{in}$ is modelled as a source term of limited spatial extent on the right hand side of Eq. (1). To clarify this point, we propose to add this source term explicitly to Eq. (1), to be corrected to:

$$\frac{\partial \tilde{z}}{\partial t} = -\nabla \cdot \mathbf{q} + Q_{in} \delta(\mathbf{x_s}) \, , \tag{1}$$

where $\delta$ is the Dirac delta function, $\mathbf{x_s} = (x_s, y_s)$ is the location of the source, and $Q_{in}$ is the inflow source volumetric flux. We propose to also explicitly add $Q_{in}$ to the corresponding numerical statement, and correct Eq. (8) to:

$$\frac{\tilde{z}_i^{\text{new}} - \tilde{z}_i}{\Delta t} = -\frac{1}{A_{CV,i}} \sum_{j=1}^{m} Q_j + \frac{Q_{in,i}}{A_{CV,i}} \, , \tag{8}$$

By this correction, Eq. (8) can match Eq. (10).

Using these equations, the simulation can start either with a given thickness distribution or with a pre-existing bed and one or more input source(s). For the simulation of each experimental case in Fig. 10 and 11, we started the simulation with a pre-existing bed and assign constant influx through a given simulation time at one or more source(s) points.

*Review Comment:*

*Figure 5: If the deposit thickness H is measured at the apex, I have some difficulties in relating the values to the legend.*

Our reply:

Thank you for flagging this error in the legend. We updated the figure with corrected legend, see below.

*Review Comment:*

*Section 7: I am not convinced that the comparison with analytical solutions should be considered so extensively. These comparisons only illustrate that the numerical implementation of the model works and have nothing to do with the applicability of the model. So the excellent agreement should not be stressed too much.*

[Figure]

**Figure 5.** Analytical solutions for the centerline profiles of cohesive-frictional deposits on an inclined plane of slope $\tan\beta = 0.02$, for different deposit heights, assuming identical material properties $\tan\phi = 0.05$, $\tau_Y/(\rho g) = 0.01$ m.

Our reply:

The importance of this comparison is to provide verification of the CVFEM model. However, in line with the review comment,
125   in the revised manuscript we will streamline discussion around this point.

*Review Comment:*

*Anyway, I enjoyed reading the manuscript and like the approach in principle, despite my criticism.*
130

Our reply:

Thank you very much for this final encouraging comment.

---

## Author Response (AR2)

**Response to editor comments**

Tzu-Yin Kasha Chen[1], Ying-Chen Wu[1], Chi-Yao Hung[2], Hervé Capart[1], and Vaughan R. Voller[3]

[1]Dept of Civil Engineering and Hydrotech Research Institute, National Taiwan University, Taiwan
[2]Dept of Soil and Water Conservation, National Chung-Hsing University, Taiwan
[3]Department of Civil, Environmental, and Geo- Engineering, University of Minnesota, USA

**Correspondence:** Tzu-Yin Kasha Chen (d06521004@ntu.edu.tw)

**Editor (Professor Andreas Lang)**

*Editor decision: Publish subject to technical corrections*

*Dear Tzu-Yin Kasha Chen and author team,*

5     *I am happy to convey that your manuscript has now been accepted for publication in ESurf subject to the technical corrections indicated by Tom Coulthard.*

    *Thank you for working with the associate editor and his reviewers. The comments have helped to improve the earlier versions of your text.*

10  *Best wishes,*

*Andreas Lang*

**Associate editor (Professor Tom Coulthard)**

15 *Associate editor decision: Publish subject to technical corrections:*

*I would like to thank the authors for their hard and considerate work revising the paper. There are a couple of fairly minor changes that I think can be taken care of under technical corrections.*

*Additional private note:*

20     *Thanks for the changes made. I have one fairly small suggestion - that I would like to see carried out under technical corrections. Reviewer 2 (Stefan Hergarten) discussed the computational efficiency of your algorithms - and you have a very clear response to this in the 'response to reviewers suggestions' pdf - lines 270-300. I know that when the paper is fully published this response will be public as part of the review package - but I think Stefans points are important and it would be helpful to the readers if a sentence or two or three covering your response could be added to the manuscript.*

25

*Many thanks,*

*Tom Coulthard*

Our reply:

We wish to thank Professor Andreas Lang and Professor Tom Coulthard for appreciating the manuscript and the constructive comments. We agree adding the response to the computational efficiency discussion is important and will strengthen our paper. We added the response (highlighted in blue) in the last paragraph in Section 7:

In Table 1, we also report the computational time in seconds needed to run these simulations on an i5-9500 Intel processor. We emphasize that the use of the dynamic time step in our solution contributes significantly to its efficiency. Preliminary versions of the code used a constant time step selected by $\Delta t = 0.25 \Delta \ell^2 / v^*$. This approach produces the same predictions as those reported here but requires over an order of magnitude more CPU time.

And in the last paragraph of the conclusion:

Despite these current limitations, we have shown that a critical slope model accounting for yield stress and friction angle can simulate deposit morphology with excellent efficiency using dynamic time steps. Aside from computation time, another key consideration is the work involved in calibrating model parameters. In this regard, an important advantage of our proposed simple model is that its parameters can be calibrated directly from topography profile data. As done in the paper for the experimental cases, all model parameters can be acquired from a single long profile through observed deposits. It is therefore not necessary to run the three-dimensional model multiple times to adjust model parameters by trial and error. More complex models, by contrast, typically require multiple iterations, or must rely on other sampling and material analysis to acquire parameter values. The model can easily calibrated parameters for a broader range of conditions than considered previously. To simulate such deposits in complex geometries, moreover, the control volume finite element method (CVFEM) was found to provide a promising numerical approach, and could possibly be extended in the future to more general processes or other geomorphic systems.